# Continuous Variable Hamiltonian Learning at Heisenberg Limit via Displacement-Random Unitary Transformation

**Xi Huang** [2]  **Lixing Zhang** [3]  **Di Luo** [1 4]

## Abstract

Characterizing continuous-variable (CV) Hamiltonians can be formulated as Hamiltonian learning under quantum measurement constraints: finite operator coefficients are inferred from noisy measurement outcomes obtained by probing an infinite-dimensional system. Existing Heisenberg-limited CV protocols are often limited to low-order structures, vulnerable to noise, or unresolved for generic multi-mode settings. We introduce Displacement-Random Unitary Transformation (D-RUT), an active data acquisition protocol with pre-specified probes and number-preserving transformations that reduce finite-order bosonic Hamiltonian learning to polynomial recovery. We prove Heisenberg-limited total evolution time with robustness to state preparation and measurement (SPAM) errors, and develop hierarchical multi-mode coefficient recovery with better statistical efficiency than simultaneous estimation. We also extend D-RUT to first-quantized Hamiltonian coefficient learning, and numerical experiments on single- and multi-mode nonlinear systems validate the predicted Heisenberg scaling.

## 1. Introduction

Precise characterization of Hamiltonians is fundamental to experimental quantum information science (Santagati et al., 2017; Guo et al., 2025; Dutt et al., 2021) and quantum computing. Describing interacting bosonic modes, CV systems are ubiquitous in quantum technologies, including quantum communication (Xu et al., 2024), networking (Stolk et al., 2024), computation (Konno et al., 2024; Lee et al., 2024; Araz et al., 2024), and metrology (Fadel et al., 2024; Kwon et al., 2022). While significant progress has been made in learning Hamiltonians for discrete systems, such as qubits (Huang et al., 2023; Bakshi et al., 2024; Hu et al., 2025) and fermions (Mirani & Hayden, 2024), the study of CV systems is often limited to structural restrictions due to infinite dimensionality of the Hilbert space. Notably, learning CV Hamiltonians imposes distinct challenges absent in discrete systems. The infinite-dimensional nature of the CV Hilbert space makes coefficient learning highly non-trivial. Moreover, higher-order terms introduce strong nonlinearities into the system dynamics, causing errors to amplify rapidly with order, creating additional challenges for the accurate estimation of the CV Hamiltonians.

Recently, achieving Heisenberg-limit scaling for CV Hamiltonian learning has become a focal point of research (Li et al., 2024; Möbus et al., 2025). However, existing protocols face significant limitations: they are typically restricted to low-order approximations, vulnerable to external noises such as state preparation and measurement (SPAM) error, or become experimentally infeasible when extended to higher-order terms (Möbus et al., 2025). Therefore, a generic protocol capable of learning arbitrary, multi-mode, but fixed finite-order bosonic operators with high experimental accessibility remains elusive.

To address these challenges, we propose an efficient framework for CV Hamiltonian learning that bridges quantum measurement constraints and statistical coefficient recovery. We introduce Displacement-Random Unitary Transformation (D-RUT), a Hamiltonian learning protocol with quantum measurement constraints. Following the statistical learning view of Hamiltonian learning, the model class is the family of finite-order CV Hamiltonians in Eqs. (1) and (2), and the unknown object is the corresponding coefficient vector, denoted by $\mathbf{h}$ locally to avoid overloading the notation used elsewhere in this paper. A training example is generated by choosing an experimentally accessible probe setting, denoted by $\mathsf{a} = (\beta, \kappa)$, together with the associated displacement and number-preserving transformations, and then measuring the ancilla. Conditioned on the probe and on the unknown coefficients, the outcome follows an explicit quantum response model. For example, displaying

[1]Department of Physics, Tsinghua University, Beijing 100084, China [2]School of Stomatology, Peking University, Beijing, 100081, China [3]Department of Chemistry and Biochemistry, University of California, Los Angeles, CA 90095, USA [4]Institute of Advanced Study, Tsinghua University, Beijing 100084, China. Correspondence to: Di Luo <diluo@tsinghua.edu.cn>.

*Proceedings of the $43^{rd}$ International Conference on Machine Learning*, Seoul, South Korea. PMLR 306, 2026. Copyright 2026 by the author(s).

*Table 1.* Terminology bridge between ML language and the Hamiltonian-learning formulation used in this work.

| ML Concept | Hamiltonian Learning (This Work) |
| --- | --- |
| Model class | Finite-order CV Hamiltonians with unknown coefficients. |
| Parameters | Hamiltonian coefficients, including single-mode, coupling, and physical position–momentum coefficients. |
| Query / input | Designed physical probe settings, primarily displacements and RPE settings. |
| Data | Quantum measurement outcomes collected at the chosen probes. |
| Response | Noisy estimates of the scalar polynomial response $C(\beta)$. |
| Objective | Accurate coefficient recovery, measured by RMSE. |
| Active data acquisition | D-RUT convert infinite-dimensional dynamics into recoverable polynomial responses rather than passively receiving a dataset. |
| Estimator | D-RUT followed by Chebyshev interpolation and Fourier inversion. |
| Resource complexity | Measurement/evolution-time complexity, with Heisenberg-limited scaling in target precision. |

the coefficient dependence as $C(\beta; \mathbf{h})$, the $X$-basis ancilla statistic satisfies

$$P_0^{\mathrm{Re}}(\mathsf{a}; \mathbf{h}) = \frac{1 + \cos(\kappa C(\beta; \mathbf{h}))}{2},$$

with an analogous sine response in the $Y$ basis. Thus, D-RUT learns a finite Hamiltonian model from measurement outcomes generated by controlled quantum queries, rather than from passively given i.i.d. samples.

Figure 1 illustrates this learning flow. The side panels show the two coefficient parameterizations learned by the protocol: bosonic coefficients in second quantization and physical coefficients in first quantization. The shared D-RUT core implements the data acquisition map: displacement and random unitary transformations convert the infinite-dimensional dynamics into scalar polynomial responses $C(\beta; \mathbf{h})$, and RPE supplies noisy observations of those responses. The classical reconstruction stage is the estimator: Chebyshev interpolation recovers the radial coefficients, Fourier inversion resolves the bosonic coefficients. Equivalently, this stage fits $\mathbf{h}$ so that the polynomial responses predicted by the Hamiltonian match the measured responses at the queried probes, with coefficient RMSE serving as the model-inference error. After recovery, the learned Hamiltonian induces predictions of $C(\beta; \widehat{\mathbf{h}})$ and hence of the corresponding ancilla measurement statistics for new probe settings in the same physical query domain. Table 1 is a terminology bridge that summarizes this mapping between statistical learning components and their quantum realization.

To address these challenges, we propose an efficient framework for CV Hamiltonian learning that bridges quantum measurement constraints and statistical coefficient recovery. Our main contributions are summarized as follows:

- **D-RUT for Heisenberg-limited CV Hamiltonian learning:** We introduce the Displacement-Random Unitary Transformation (D-RUT) protocol, an active data acquisition and structured coefficient-recovery method that learns generic bosonic Hamiltonians with Heisenberg-limited precision ($T \sim \mathcal{O}(1/\epsilon)$). The protocol uses designed physical probes to expose recoverable polynomial responses and employs a hierarchical recovery strategy that provably improves statistical efficiency for multi-mode systems with multiple bosonic degrees of freedom (DOFs) compared to prior art (Figure 1).

- **Application to first and second quantization:** Beyond the standard second-quantized bosonic setting, our protocol also applies naturally in the first-quantized regime, enabling the learning of physical Hamiltonian coefficients expressed directly in position and momentum operators with Heisenberg-limited precision. This is achieved by reformulating the learning problem within a new bosonic basis defined by a known reference frame; the physical coefficients are recovered through direct linear inversion, provided the frame mismatch is bounded.

- **Robustness guarantee:** Compared with previous work (Möbus et al., 2025), we establish theoretical guarantees for the error tolerance of D-RUT, specifically for state preparation and measurement (SPAM) errors. In addition, D-RUT only requires access to the vacuum state and displacement operators, which maintains high experimental accessibility.

## 2. Related Work

For an unknown Hamiltonian, the most intuitive approach to estimating a coefficient with precision $\epsilon$ is by ensemble averaging over measurements. However, by the central limit theorem, the total evolution time $T$ would scale as $\mathcal{O}(\epsilon^{-2})$. This scaling is known as the standard quantum limit (SQL). By exploiting quantum resources such as entanglement and coherent control, estimation protocols surpassing the SQL have been proposed (Huelga et al., 1997; Escher et al., 2011; Pezze et al., 2018). Ultimately, the fundamental precision bound imposed by quantum mechanics is the Heisenberg

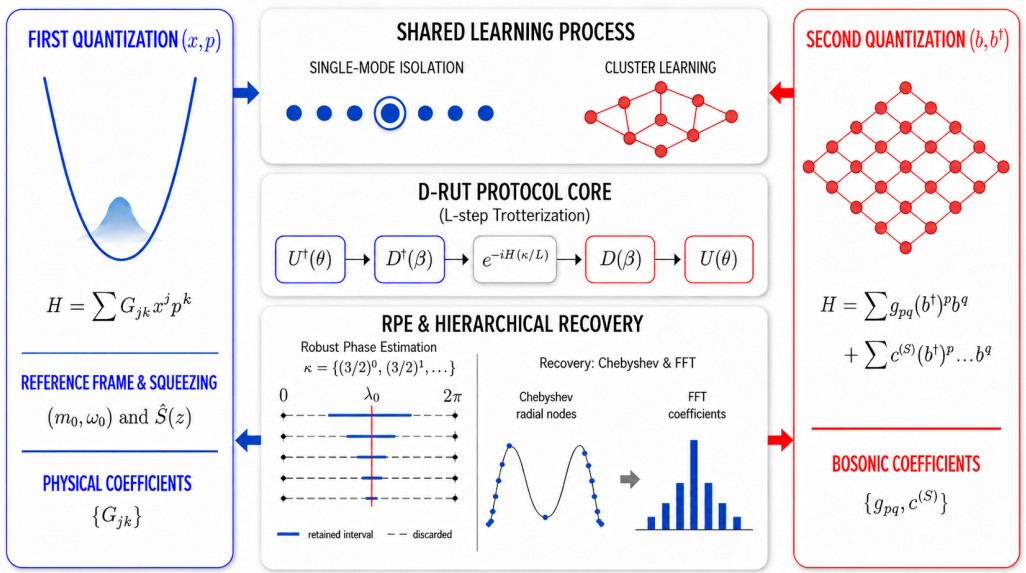

*Figure 1.* D-RUT-based learning pipeline for second-quantized bosonic coefficients and first-quantized physical coefficients. Displacement and random unitary transformations implement controlled feature construction, robust phase estimation obtains noisy scalar responses $C(\beta)$, and classical inversion recovers the coefficient vector.

limit, $T \sim \mathcal{O}(1/\epsilon)$, which arises from the Heisenberg uncertainty principle. Recently, Heisenberg-limited Hamiltonian learning has been achieved in several settings, including qubit systems implementable on quantum circuits (Huang et al., 2023; Bakshi et al., 2024), fermionic systems such as Hubbard models (Mirani & Hayden, 2024), and light–matter hybrid systems applicable to characterization of non-Markovian noises (Zhang et al., 2025).

On the other hand, neural-network-based approaches have also been explored for Hamiltonian learning (Han et al., 2021; Liu & Wang, 2025). By training physics-informed neural networks on time-series measurement data generated by an unknown Hamiltonian, these methods can approximate the underlying Hamiltonian and reproduce the system dynamics over a finite evolution time. However, such approaches typically lack rigorous error bounds and do not provide guarantees on precision scaling. Moreover, their applicability to continuous-variable systems, particularly those governed by strongly nonlinear Hamiltonians with higher-order terms, remains limited.

For CV systems, Heisenberg-limited estimation can in principle be achieved using squeezed quantum states (Zhuang et al., 2018). Despite their high parallelizability, such schemes are particularly vulnerable to external noise and experimental imperfections. Alternative approaches based on engineered dissipation (Möbus et al., 2025) and random unitary transformations (Li et al., 2024) have also been proposed to achieve Heisenberg scaling. However, the former does not provide provable robustness guarantees against experimental noise, while the latter relies on prior assumptions

about the specific low-order structures of the Hamiltonian operators.

To address these limitations, we propose the D-RUT algorithm, which enables the estimation of higher-order, multi-mode continuous-variable Hamiltonians with provably bounded error tolerance. In ML terms, the displacement parameter is a designed input, the D-RUT is a physically implemented feature construction, RPE supplies noisy responses $C(\beta)$, and Chebyshev/Fourier inversion is the closed-form estimator for the Hamiltonian coefficient vector.

## 3. Preliminary

We consider a CV system composed of multi bosonic modes, where each mode is associated with an infinite-dimensional Hilbert space known as the Fock space. The interactions within this system are described by unbounded operators, expressed either via creation $\hat{b}^\dagger$ and annihilation $\hat{b}$ operators (second quantization) or position $\hat{x}$ and momentum $\hat{p}$ operators (first quantization).

We first address the learning of a generic high-order bosonic Hamiltonian involving $N$ modes. The Hamiltonian is defined as a linear combination of creation and annihilation

operators raised to non-negative integer powers ($p, q \in \mathbb{N}_0$):

$$\hat{H} = \sum_{\zeta=1}^{N} \sum_{\substack{(p_\zeta, q_\zeta) \\ p_\zeta + q_\zeta \leq d}} g_{p_\zeta, q_\zeta}^{(\zeta)} (\hat{b}_\zeta^\dagger)^{p_\zeta} \hat{b}_\zeta^{q_\zeta}$$
$$+ \sum_{\substack{S \subseteq \{1,...,N\} \\ |S| \geq 2}} \sum_{\substack{(\mathbf{p}_S, \mathbf{q}_S) \\ 0 < \|\mathbf{p}_S\|_1 + \|\mathbf{q}_S\|_1 \leq d}} c_{\mathbf{p}_S, \mathbf{q}_S}^{(S)} (\hat{b}_S^\dagger)^{\mathbf{p}_S} (\hat{b}_S)^{\mathbf{q}_S}, \tag{1}$$

where $\hat{b}_\zeta^\dagger$ and $\hat{b}_\zeta$ denote the creation (annihilation) operators for the $\zeta^{th}$ bosonic mode. Here, $g_{p_\zeta, q_\zeta}^{(\zeta)}$ represents the single-mode on-site coefficient, and $c_{\mathbf{p}_S, \mathbf{q}_S}^{(S)}$ represents the multi-mode coupling coefficient. For brevity, we index the modes in each interaction term using an ordered set $S = \{s_1, s_2, \ldots, s_{|S|}\}$. Accordingly, $\mathbf{p}_S$ and $\mathbf{q}_S$ are tuples specifying the powers of $\hat{b}_\zeta^\dagger$ and $\hat{b}_\zeta$, with $\mathbf{p}_S \equiv (p_{s_1}, p_{s_2}, \ldots, p_{s_{|S|}})$ and $(\hat{b}_S^\dagger)^{\mathbf{p}_S} = \prod_i^{|S|} (\hat{b}_{s_i}^\dagger)^{p_{s_i}}$ (similarly for $\mathbf{q}_S$). This formulation encompasses all possible combinations of single and multi-mode terms up to order $d$, representing an exceptionally general model class.

We further generalize our framework to the first-quantized regime, enabling the learning of physical Hamiltonians expressed in position and momentum operators. Similar to the bosonic case, a generic $N$-mode Hamiltonian is defined as a symmetrized polynomial of physical operators $\{\hat{x}_\zeta, \hat{p}_\zeta\}_{\zeta=1}^{N}$:

$$\hat{H} = \sum_{\zeta=1}^{N} \sum_{\substack{(j,k) \\ 0 < j+k \leq d}} G_{j,k}^{(\zeta)} \{\hat{x}_\zeta^j \hat{p}_\zeta^k\}_S$$
$$+ \sum_{\substack{S \subseteq \{1,...,N\} \\ |S| \geq 2}} \sum_{\substack{(\mathbf{j}_S, \mathbf{k}_S) \\ 0 < \|\mathbf{j}_S\|_1 + \|\mathbf{k}_S\|_1 \leq d}} G_{\mathbf{j}_S, \mathbf{k}_S}^{(S)} \prod_{\zeta \in S} \{\hat{x}_\zeta^{j_\zeta} \hat{p}_\zeta^{k_\zeta}\}_S, \tag{2}$$

where $G_{j,k}^{(\zeta)}$ and $G_{\mathbf{j}_S, \mathbf{k}_S}^{(S)}$ are the real physical coefficients to be learned. The symmetrization is applied within each mode as $\{\hat{x}^j \hat{p}^k\}_S := \frac{1}{2}(\hat{x}^j \hat{p}^k + \hat{p}^k \hat{x}^j)$. Our protocol expresses $\hat{H}$ in the normal-ordered basis of a set of new bosonic operators $\{\hat{B}_\zeta, \hat{B}_\zeta^\dagger\}$ defined by a known reference frame $(m_0, \omega_0)$. The dimensionless operators are given by $\hat{X}_\zeta = \sqrt{m_0 \omega_0} \hat{x}_\zeta$ and $\hat{P}_\zeta = \frac{1}{\sqrt{m_0 \omega_0}} \hat{p}_\zeta$. Analogous to the second-quantized case, to maintain brevity, the first-quantized protocol in the subsequent sections will focus on the single-mode Hamiltonian form $\hat{H} = \sum G_{j,k} \{\hat{x}^j \hat{p}^k\}_S$, while the extension to multi-mode follows the framework in Section 6.

The task of Hamiltonian learning is to estimate the unknown coefficients (e.g., $\{g^{(\zeta)}, c^{(S)}\}$ or $\{G^{(\zeta)}, G^{(S)}\}$) given black-box access to the unitary evolution $e^{-i\hat{H}t}$. A fundamental benchmark in this domain is the Heisenberg limit, which

requires the estimation error $\epsilon$ to scale inversely with the total evolution time, i.e., $T \sim \mathcal{O}(1/\epsilon)$.

From the statistical-learning perspective summarized in Table 1, the protocol should be read as a designed data-acquisition and coefficient-recovery procedure under a quantum measurement oracle. In the following section, we present our main results establishing protocols that achieve Heisenberg-limited scaling for these general Hamiltonian classes.

## 4. Main Results

We present a framework for learning generic high-order Hamiltonians in both first and second quantization. For any bosonic Hamiltonian satisfying the general form in Eq. 1, our protocol guarantees the following:

**Theorem 4.1.** *Given unitary access to a generic multi-mode bosonic Hamiltonian in the form of Eq. 1, there exists a learning protocol that estimates all Hamiltonian coefficients up to a Root-Mean-Square Error (RMSE) $\epsilon$, satisfying:*

1. **Heisenberg-Limited Scaling:** *The protocol requires a total evolution time of $T \sim \mathcal{O}(\epsilon^{-1})$.*

2. **Statistical Efficiency:** *The protocol utilizes a hierarchical recovery scheme that achieves a lower estimation error bound compared to the simultaneous recovery scheme in (Möbus et al., 2025).*

3. **Robustness:** *The estimation remains robust against bounded SPAM errors.*

To establish Theorem 4.1, we propose the Displacement-Random Unitary Transformation (D-RUT) protocol. The key insight is to map the target Hamiltonian into a number-conserving effective operator $\hat{\mathcal{H}}(\beta)$ by averaging the displaced dynamics over random unitary rotations. The eigenvalues of $\hat{\mathcal{H}}(\beta)$ encode the target coefficients into a measurable constant term $C(\beta) = \sum_{0 < p+q \leq d} g_{p,q} (\beta^*)^p \beta^q$, which is estimated using Robust Phase Estimation (RPE) (Kimmel et al., 2015; Ni et al., 2023) with Heisenberg-limited scaling and recovered by Chebyshev interpolation and discrete Fourier transformation (Möbus et al., 2025). In learning terminology, each designed probe $\beta$ maps the unknown Hamiltonian coefficients to a scalar polynomial response $C(\beta)$, while RPE supplies a noisy estimate of this response with Heisenberg-limited cost. For a multi-mode system, we use the "divide-and-conquer" strategy to decouple it as a series of $N$-mode systems ($N \sim \mathcal{O}(1)$) that can be learned in parallel. For each of them, we selectively zero out displacement parameters to decouple the interaction clusters, allowing for a hierarchical recovery of coefficients: first learning the pure single-mode coefficients, and subsequently resolving the single-mode and multi-mode coupling

coefficients within cluster $S$. We then develop a protocol for learning the first-quantized Hamiltonian, enabling the learning of physical parameters for Hamiltonians expressed in position and momentum operators, with the following guarantees:

**Theorem 4.2.** *Given unitary access to a generic first-quantized Hamiltonian in the form of Eq. 2, there exists a protocol that learns all physical coefficients $\{G^{(\zeta)}, G^{(S)}\}$ up to a Root-Mean-Square Error (RMSE) $\epsilon_G$, provided that the mismatch between the reference frame $(m_0, \omega_0)$ and the physical parameters $(m, \omega)$ is bounded. The protocol satisfies:*

1. **Heisenberg-Limited Scaling:** *The total evolution time scales as $T \sim \mathcal{O}(\epsilon_G^{-1})$.*

2. **Robustness:** *The estimation remains robust under small SPAM errors.*

In Table 2, we summarize the features of our work and compare them with state-of-the-art methods in CV Hamiltonian learning. While all of these works achieve the Heisenberg limit, (Li et al., 2024) is restricted to low-order operator approximations, and (Möbus et al., 2025) is sensitive to state-preparation-and-measurement (SPAM) errors. Neither work addresses the learning of Hamiltonians formulated in first quantization.

*Table 2.* Comparison of provable features of Heisenberg-limited CV Hamiltonian-learning protocols.

| Feature | (Li et al., 2024) | (Möbus et al., 2025) | **Ours** |
|---|---|---|---|
| Heisen. limit | ✓ | ✓ | ✓ |
| Higher-order | ✗ | ✓ | ✓ |
| SPAM-robust | ✓ | ✗ | ✓ |
| 1st-quant. | ✗ | ✗ | ✓ |

# 5. Learning a Single-Mode Hamiltonian via D-RUT

In this section, we detail the protocol for learning the coefficients of a general single-mode bosonic Hamiltonian, $\hat{H} = \sum_{0 < p+q \le d} g_{p,q}(\hat{b}^\dagger)^p \hat{b}^q$. Our strategy transforms the high-dimensional coefficient estimation problem into a series of phase-estimation tasks via the Displacement-Random Unitary Transformation (D-RUT). From the statistical-learning viewpoint, the displacement $\beta$ is the designed input, the vacuum eigenvalue $C(\beta)$ is the scalar response estimated from noisy measurements, and the final Chebyshev/Fourier recovery step is a structured closed-form regression estimator. The protocol proceeds in three main steps: first, we apply a displacement operator $\hat{D}(\beta)$ followed by random phase rotations to eliminate non-number-conserving terms. The

---

**Algorithm 1** Learning of single-mode bosonic coefficients.

**Input:** Unknown Hamiltonian $\hat{H}$, maximum order $d$, target precision $\epsilon$.
**Output:** Estimated coefficients $\{\hat{c}_{p,q}\}_{0 < p+q \le d}$.
Define $d + 1$ radial Chebyshev nodes $\{r_\mu\}$ on $[r_{\min}, r_{\max}]$.
Define sampling angles $\Theta = \{\theta_{u,l} = \frac{\pi u}{l+1} \mid 1 \le l \le d, 0 \le u \le l\}$.
**for** each angle $\theta \in \Theta$ **do**
    **for** $\mu = 1$ to $d + 1$ **do**
        Set displacement parameter $\beta = r_\mu e^{i\theta}$.
        **State Preparation:** Initialize ancilla in $\frac{1}{\sqrt{2}}(|0\rangle_{\text{anc}} + |1\rangle_{\text{anc}})$ and system in $|\text{vac}\rangle$.
        **Evolution:** Apply the ancilla-controlled unitary $\hat{\mathcal{U}}(\kappa)$ for RPE iteration $\kappa$, where $\hat{\mathcal{U}}(\kappa)$ is an $L$-step Trotterized D-RUT sequence:
        $\hat{\mathcal{U}}(\kappa) = \prod_{j=1}^{L} \left[ \mathbf{U}^\dagger(\theta_j)\hat{D}^\dagger(\beta)e^{-i\hat{H}(\kappa/L)}\hat{D}(\beta)\mathbf{U}(\theta_j) \right]$.
        **Measurement:** Measure ancilla in X/Y bases to estimate the phase $C(r_\mu, \theta)$ via RPE up to precision $\epsilon$.
    **end for**
    **Radial Inversion:** Solve for intermediate coefficients $\{g_l(\theta)\}_{l=1}^d$ using Chebyshev interpolation on the collected values $\{C(r_\mu, \theta)\}_{\mu=1}^{d+1}$.
**end for**
**for** $l = 1$ to $d$ **do**
    Collect values $\{g_l(\theta_{u,l})\}_{u=0}^l$.
    **Angular Inversion:** Recover final coefficients $\{g_{p,q}\}_{p+q=l}$ via inverse discrete Fourier transform.
**end for**
**Return** coefficients $\{g_{p,q}\}$.

---

vacuum eigenvalue of the effective Hamiltonian, denoted as $C(\beta)$, encodes the target coefficients in a polynomial form. Second, we estimate $C(\beta)$ with Heisenberg-limited precision using Robust Phase Estimation (RPE), implemented via a Trotterized sequence of D-RUT operations controlled by an ancilla qubit. Finally, by varying the displacement parameter $\beta$, we recover the individual coefficients $\{g_{p,q}\}$ through a coefficient recovery strategy involving Chebyshev interpolation and discrete Fourier transform.

## 5.1. The D-RUT Method

The Displacement-Random Unitary Transformation (D-RUT) serves as the core subroutine of our protocol. It is a two-step procedure designed to project a general bosonic Hamiltonian into a number-conserving effective operator.

We first subject the original Hamiltonian $\hat{H}$ to a displacement operator $\hat{D}(\beta) = e^{\beta \hat{b}^\dagger - \beta^* \hat{b}}$. This transformation co-

herently shifts the creation and annihilation operators:

$$\hat{D}^\dagger(\beta)\hat{b}\hat{D}(\beta) = \hat{b} + \beta, \ \hat{D}^\dagger(\beta)\hat{b}^\dagger\hat{D}(\beta) = \hat{b}^\dagger + \beta^*, \quad (3)$$

where $\beta \in \mathbb{C}$ is a tunable complex displacement parameter. This operation yields the displaced Hamiltonian $\hat{H}_D(\beta) = \hat{D}^\dagger(\beta)\hat{H}\hat{D}(\beta)$.

Subsequently, we eliminate non-number-conserving terms by averaging $\hat{H}_D(\beta)$ over a group of random phase rotations, a technique known as Random Unitary Transformation (RUT) (Zhang et al., 2025; Li et al., 2024). We define the effective Hamiltonian, $\hat{\mathcal{H}}(\beta)$, as the expectation over $\mathbf{U}(\theta) = e^{-i\theta\hat{N}}$ where $\hat{N} = \hat{b}^\dagger\hat{b}$:

$$\hat{\mathcal{H}}(\beta) = \mathbb{E}_{\theta \sim \mathcal{U}[0,2\pi]}[\mathbf{U}^\dagger(\theta)\hat{H}_D(\beta)\mathbf{U}(\theta)] \quad (4)$$

$$= \frac{1}{2\pi}\int_0^{2\pi} d\theta \, \mathbf{U}^\dagger(\theta)\hat{H}_D(\beta)\mathbf{U}(\theta),$$

where $\mathcal{U}[0, 2\pi]$ denotes the uniform distribution. This transformation acts as a projector onto the diagonal basis of the number operator. Specifically, for any monomial term $(\hat{b}^\dagger)^p\hat{b}^q$, the expectation vanishes unless the number of creation and annihilation operators are equal ($p = q$):

$$\mathbb{E}_{\theta \sim \mathcal{U}[0,2\pi]}[\mathbf{U}^\dagger(\theta)(\hat{b}^\dagger)^p\hat{b}^q\mathbf{U}(\theta)] \quad (5)$$

$$= (\hat{b}^\dagger)^p\hat{b}^q\frac{1}{2\pi}\int_0^{2\pi} e^{i(p-q)\theta}d\theta = (\hat{b}^\dagger)^p\hat{b}^q\delta_{pq}.$$

Consequently, the ideal effective Hamiltonian $\hat{\mathcal{H}}(\beta)$ reduces to a polynomial in the number operator $\hat{N}$:

$$\hat{\mathcal{H}}(\beta) = d_k(\beta)\hat{N}^k + \cdots + d_1(\beta)\hat{N} + C(\beta), \quad (6)$$

where the constant term $C(\beta)$ is the linear combination of the displacement parameter $\beta$.

## 5.2. Measurement via Robust Phase Estimation

To extract $C(\beta)$, we employ the RPE protocol (Kimmel et al., 2015; Ni et al., 2023; Möbus et al., 2025). We construct the unitary evolution required for RPE by Trotterizing the ideal dynamics defined in Eq. 4. The sequence $\hat{\mathcal{U}}(\kappa)$ is given by:

$$\hat{\mathcal{U}}(\kappa) = \prod_{j=1}^{L}\left[\mathbf{U}^\dagger(\theta_j)\hat{D}^\dagger(\beta)e^{-i\hat{H}(\kappa/L)}\hat{D}(\beta)\mathbf{U}(\theta_j)\right], \quad (7)$$

where $L$ is the number of gates and each $\theta_j$ is independently sampled from $\mathcal{U}[0, 2\pi]$. In the limit $L \to \infty$, the action of $\hat{\mathcal{U}}(\kappa)$ on the bosonic vacuum state $|\text{vac}\rangle$ converges to the ideal phase evolution generated by $\hat{\mathcal{H}}(\beta)$, as illustrated in Figure 2:

$$\hat{\mathcal{U}}(\kappa)|\text{vac}\rangle \approx e^{-i\hat{\mathcal{H}}(\beta)\kappa}|\text{vac}\rangle = e^{-iC(\beta)\kappa}|\text{vac}\rangle. \quad (8)$$

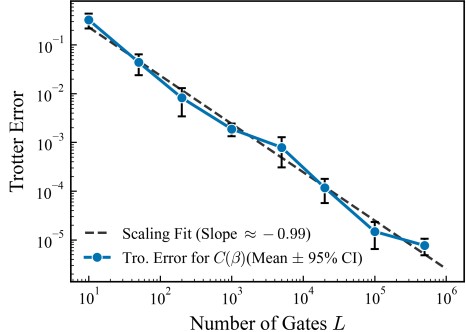

*Figure 2.* Convergence of the estimation of $C(\beta)$ with increasing $L$.

The phase estimation circuit initializes an ancilla-system state $|\psi_0\rangle = |0\rangle_{\text{anc}} \otimes |\text{vac}\rangle$. A Hadamard gate creates the superposition $\frac{1}{\sqrt{2}}(|0\rangle_{\text{anc}} + |1\rangle_{\text{anc}}) \otimes |\text{vac}\rangle$. Applying the controlled-$\hat{\mathcal{U}}(\kappa)$ operation yields the final state:

$$|\psi_\kappa\rangle = \frac{1}{\sqrt{2}}\left(|0\rangle_{\text{anc}} \otimes |\text{vac}\rangle + |1\rangle_{\text{anc}} \otimes \hat{\mathcal{U}}(\kappa)|\text{vac}\rangle\right) \quad (9)$$

$$= \frac{1}{\sqrt{2}}\left(|0\rangle_{\text{anc}} + e^{-iC(\beta)\kappa}|1\rangle_{\text{anc}}\right) \otimes |\text{vac}\rangle. \quad (10)$$

Projective measurements on the ancilla in the X and Y bases provide the statistics necessary for RPE. Specifically, the probability of observing $|0\rangle_{\text{anc}}$ in the $X/Y$ basis is:

$$P_0^{\text{Re}} = \frac{1 + \cos(\kappa C(\beta))}{2}, \ P_0^{\text{Im}} = \frac{1 + \sin(\kappa C(\beta))}{2}. \quad (11)$$

Within the RPE subroutine, selecting $\kappa$ from the geometric sequence $\{(3/2)^0, (3/2)^1, \ldots, (3/2)^K\}$ and iteratively narrowing the confidence interval (Möbus et al., 2025) estimates $C(\beta)$ with total evolution time scaling at the Heisenberg limit. Importantly, while finite $L$ introduces systematic Trotterization error, the Heisenberg scaling of the total evolution time is strictly preserved, provided that $L$ is chosen sufficiently large to suppress the trotter error of the longest single-shot evolution below the RPE tolerance threshold.

## 5.3. Coefficient Recovery Strategy

With the ability to estimate $C(\beta)$ for any displacement $\beta = re^{i\theta}$, we recover the unknown coefficients $\{g_{p,q}\}$ using a two-stage strategy (Möbus et al., 2025). We decompose the constant term into radial and angular components: $C(r, \theta) = \sum_{l=1}^{d} r^l g_l(\theta)$, where the intermediate coefficient $g_l(\theta) := \sum_{p+q=l} g_{p,q}e^{i(q-p)\theta}$ acts as a Fourier series for terms of order $l$.

To extract $\{g_l(\theta)\}$, we perform RPE experiments at $d + 1$ Chebyshev radial nodes $\{r_\mu\}$ to solve the polynomial fitting problem. This choice minimizes the condition number of the

Vandermonde matrix, suppressing the Runge phenomenon. Subsequently, to recover the individual coefficients $g_{p,q}$, we sample $g_l(\theta)$ at $l+1$ uniform angles and apply the inverse discrete Fourier transform:

$$g_{p,l-p} = \frac{1}{l+1} \sum_{u=0}^{l} e^{-il\theta_{u,l}} g_l(\theta_{u,l}) e^{i\frac{2\pi pu}{l+1}}, \qquad (12)$$

where $\theta_{u,l} = \frac{\pi u}{l+1}$. The resilience of this recovery strategy against statistical noise and SPAM errors is analyzed in Appendix B and Appendix C. The complete procedure of D-RUT protocol is summarized in Algorithm 1.

## 6. Learning Multi-Mode Hamiltonians

We adopt a "divide-and-conquer" strategy (Li et al., 2024; Möbus et al., 2025) to decouple large-scale systems into independent $N$-mode subsystems ($N \sim \mathcal{O}(1)$), which are learned in parallel. Unlike simultaneous estimation on the full-dimensional hypercube, our approach utilizes the physical control of displacements to hierarchically resolve coefficients on lower-dimensional domains. In statistical learning terms, this exploits local support structure in a high-dimensional coefficient model. The total constant term $C_{\text{total}}$ derived from D-RUT is:

$$C_{\text{total}}(\boldsymbol{\beta}) = \sum_{\substack{(p_\zeta, q_\zeta) \\ p_\zeta + q_\zeta \leq d}} g_{p,q}^{(\zeta)} (\beta_\zeta^*)^{p_\zeta} \beta_\zeta^{q_\zeta} \qquad (13)$$
$$+ \sum_{\substack{(\mathbf{p}_S, \mathbf{q}_S) \\ 0 < \|\mathbf{p}_S\|_1 + \|\mathbf{q}_S\|_1 \leq d}} c_{\mathbf{p}_S, \mathbf{q}_S}^{(S)} \prod_{s_i \in S} (\beta_{s_i}^*)^{p_{s_i}} \beta_{s_i}^{q_{s_i}}.$$

We first isolate and learn the pure single-mode terms by setting the displacement parameters of other modes to zero. Then, we learn coefficients supported on cluster $S$ (including both the single-mode and the coupling terms) with a high-dimensional recovery strategy in (Möbus et al., 2025). We demonstrate a statistical advantage: this approach yields an estimation error bound that scales only with the local interaction order $|S|$ rather than the total system size $N$.

### 6.1. Hierarchical Recovery Strategy

#### 6.1.1. STEP 1: ISOLATION OF SINGLE-MODE TERMS

We first characterize the pure single-mode coefficients $\{g_{p,q}^{(\zeta)}\}$ for each mode $\zeta$ independently. By setting $\beta_\eta = 0$ for all $\eta \neq \zeta$, we physically suppress all coupling terms involving other modes. The total constant term thus collapses to a pure single-mode expression:

$$C_{\text{total}}(\boldsymbol{\beta}) = C_\zeta(\beta_\zeta) = \sum_{\substack{(p_\zeta, q_\zeta) \\ p_\zeta + q_\zeta \leq d}} g_{p,q}^{(\zeta)} (\beta_\zeta^*)^{p_\zeta} \beta_\zeta^{q_\zeta}. \qquad (14)$$

---

**Algorithm 2** Learning of single-mode physical parameters.

**Input:** Unknown Hamiltonian $\hat{H}$, max order $d$, reference frame $(m_0, \omega_0)$, squeezing $R'$.
**Output:** Physical coefficients $\{G_{j,k}\}$.
Construct the mapping matrix $\mathbf{M}$ via Eq. 20.
Define $d+1$ radial Chebyshev nodes $\{r_\mu\}$ and sampling angles $\Theta$.
**for** each angle $\theta \in \Theta$ **do**
  **for** $\mu = 1$ to $d+1$ **do**
    Set displacement $\beta = r_\mu e^{i\theta}$.
    **Reference Basis Set:** Implement $\hat{D}_{B'}(\beta) = \hat{S}^\dagger(R')\hat{D}_B(\beta)\hat{S}(R')$ and $\mathbf{U}(\theta) = e^{-i\theta\hat{N}_{B'}}$.
    **State Preparation:** Initialize ancilla in $\frac{1}{\sqrt{2}}(|0\rangle_{\text{anc}} + |1\rangle_{\text{anc}})$ and system in $|\text{vac}\rangle$.
    **Evolution:** Apply the ancilla-controlled unitary $\hat{\mathcal{U}}(\kappa)$ for RPE iteration $\kappa$ ($L$-step Trotterized D-RUT):
    $\hat{\mathcal{U}}(\kappa) = \prod_{j=1}^{L} \left[ \mathbf{U}^\dagger(\theta_j)\hat{D}_{B'}^\dagger(\beta) e^{-i\hat{H}(\kappa/L)} \hat{D}_{B'}(\beta)\mathbf{U}(\theta_j) \right]$.
    **Measurement:** Measure ancilla in X/Y bases to estimate $C(r_\mu, \theta)$ via RPE.
  **end for**
  **Radial Inversion:** Solve for intermediate coefficients $\{g_l(\theta)\}_{l=0}^{d}$ (including $l=0$) using Chebyshev interpolation.
**end for**
**Angular Inversion:** Recover measurable coefficients $\hat{\mathbf{g}}' = \{g_{p,q}'\}$ via inverse discrete Fourier transform.
**Physical Recovery:** Solve linear system $\hat{\mathbf{G}} = \mathbf{M}^{-1}\hat{\mathbf{g}}'$ using least-squares.
**Return** coefficients $\{G_{j,k}\}$.

---

This effectively reduces the problem to independent single-mode tasks, solvable via the protocol in Section 5 with optimal conditioning.

#### 6.1.2. STEP 2: LEARNING OF INTERACTION CLUSTERS

To learn all coefficients supported on cluster $S$ (including both the single-mode terms and the coupling terms), we restrict the measurements to the $|S|$-dimensional sub-manifold defined by $\beta_\nu = 0$ for all $\nu \notin S$. On this sub-manifold, the constant term becomes a polynomial for non-zero $\{\beta_\zeta\}_{\zeta \in S}$:

$$C_S(\boldsymbol{\beta}_S) = \sum_{\zeta \in S} C_\zeta(\beta_\zeta)$$
$$+ \sum_{\substack{S' \subseteq S \\ |S'| \geq 2}} \sum_{\mathbf{p}, \mathbf{q}} c_{\mathbf{p}, \mathbf{q}}^{(S')} \prod_{s_i \in S'} (\beta_{s_i}^*)^{p_{s_i}} \beta_{s_i}^{q_{s_i}}. \qquad (15)$$

We perform multi-dimensional Chebyshev interpolation and inverse Fourier transform on this $|S|$-dimensional domain without interference from the bystander $N - |S|$ modes. As proven in Appendix D, this yields superior statistical

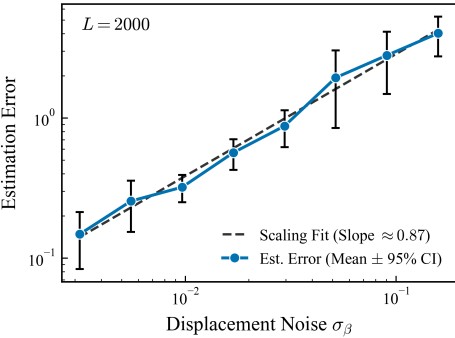

*Figure 3.* Numerical verification of robustness against SPAM error with standard deviation $\sigma_\beta$ varying across the range $[3 \times 10^{-3}, 0.15]$. For each noise level, the RMSE was averaged over 10 independent trials.

efficiency compared to simultaneous recovery strategies (Möbus et al., 2025). In summary, the complete multimode protocol is executed by iteratively applying the D-RUT procedures defined in Algorithm 1: first on isolated modes, and subsequently on interaction clusters to resolve coupling coefficients.

# 7. Learning First-Quantized Hamiltonians

While the D-RUT framework naturally operates within second quantization, we extend the protocol to the first-quantized setting to extract fundamental physical parameters associated with dimensionless position and momentum operators, $\{\hat{X}, \hat{P}\}$. These operators are defined relative to a known reference frame $(m_0, \omega_0)$, where $\hat{X} = \sqrt{m_0 \omega_0}\hat{x}$ and $\hat{P} = \frac{1}{\sqrt{m_0 \omega_0}}\hat{p}$. Without loss of generality, we consider a single-mode Hamiltonian in symmetrized form:

$$\hat{H} = \sum_{\substack{j,k \geq 0 \\ 0 < j+k \leq d}} G_{j,k}\{\hat{x}^j \hat{p}^k\}_S = \sum_{\substack{j,k \geq 0 \\ 0 < j+k \leq d}} G'_{j,k}\{\hat{X}^j \hat{P}^k\}_S,$$
(16)

where the symmetrization is defined as $\{\hat{A}^j \hat{B}^k\}_S := \frac{1}{2}(\hat{A}^j \hat{B}^k + \hat{B}^k \hat{A}^j)$. Here, $\{G_{j,k}\}$ represent the target physical coefficients, while $\{G'_{j,k}\}$ are the rescaled coefficients under the chosen reference frame.

We define a fixed reference basis of creation and annihilation operators: $\hat{B} = \frac{1}{\sqrt{2\hbar}}(\hat{X} + i\hat{P})$ and $\hat{B}^\dagger = \frac{1}{\sqrt{2\hbar}}(\hat{X} - i\hat{P})$. However, direct measurement in this fixed basis may yield numerical instabilities if the reference frame $(m_0, \omega_0)$ significantly deviates from the system's intrinsic parameters $(m, \omega)$. To address this challenge, we perform D-RUT in a tunable basis $\{\hat{B}', \hat{B}'^\dagger\}$, related to the reference basis via a squeezing operation $\hat{S}(R')$:

$$\hat{B}' = \hat{S}^\dagger(R')\hat{B}\hat{S}(R') = \hat{B}\cosh(R') - \hat{B}^\dagger \sinh(R'),$$
(17)

$$\hat{B}'^\dagger = \hat{S}^\dagger(R')\hat{B}^\dagger \hat{S}(R') = \hat{B}^\dagger \cosh(R') - \hat{B}\sinh(R').$$
(18)

Here, $\hat{S}(z) = \exp[\frac{1}{2}(z^* \hat{b}^2 - z\hat{b}^{\dagger 2})]$, and $R'$ serves as a controllable parameter to optimize the basis alignment.

Implementing D-RUT in the $\{\hat{B}', \hat{B}'^\dagger\}$ basis requires a transformed displacement operator $\hat{D}_{B'}(\beta)$. Mathematically, this relates to the reference displacement $\hat{D}_B(\beta)$ via the squeezing transformation:

$$\hat{D}_{B'}(\beta) = \exp(\beta \hat{B}'^\dagger - \beta^* \hat{B}') = \hat{S}^\dagger(R')\hat{D}_B(\beta)\hat{S}(R').$$
(19)

This equivalence implies that $\hat{D}_{B'}(\beta)$ can be physically realized as a squeezed displacement. Furthermore, the corresponding random unitary rotation $e^{-i\theta \hat{N}_{B'}}$ constitutes a Gaussian operation, as $\hat{N}_{B'}$ is quadratic in $\{B', B'^\dagger\}$. According to the Bloch-Messiah decomposition, any Gaussian transformation can be decomposed into passive linear-optical circuits and single-mode squeezing. Therefore, the D-RUT operations in the tunable basis can be implemented via (Chakhmakhchyan & Cerf, 2018).

Within this tunable basis, the position and momentum operators are expressed as:

$$\hat{X} = \sqrt{\frac{\hbar}{2}}e^{R'}(\hat{B}' + \hat{B}'^\dagger), \quad \hat{P} = i\sqrt{\frac{\hbar}{2}}e^{-R'}(\hat{B}'^\dagger - \hat{B}').$$
(20)

Consequently, any symmetrized physical operator term $\{\hat{x}^j \hat{p}^k\}_S$ admits a unique expansion into normal-ordered measurement operators $(\hat{B}'^\dagger)^p \hat{B}'^q$, $\{\hat{x}^j \hat{p}^k\}_S = \sum_{p,q} M_{pq,jk}(\hat{B}'^\dagger)^p \hat{B}'^q$, where $\mathbf{M}$ is a coefficient mapping matrix governing the basis change. Substituting this expansion into Eq. 16, we rewrite the Hamiltonian in second quantization form as $\hat{H} = \sum_{p,q} g'_{p,q}(\hat{B}'^\dagger)^p \hat{B}'^q$. The

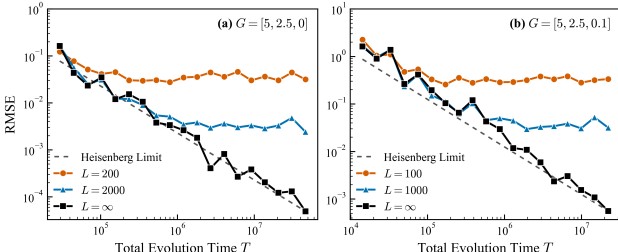

*Figure 4.* Verification of Heisenberg scaling for first-quantized Hamiltonian learning: (Left) harmonic oscillator and (Right) anharmonic oscillator. As $L$ increases, the performance converges to the limit $L \to \infty$, aligning with the Heisenberg-limit slope of $-1$.

measurable coefficients $\{g'_{p,q}\}$ are linearly related to the physical parameters $\{G_{j,k}\}$:

$$g'_{p,q} = \sum_{j,k} M_{pq,jk} G_{j,k}. \qquad (21)$$

We note that the basis transformation typically induces a constant offset $g'_{0,0}$. Our protocol can recover this term by including $l = 0$ in the radial interpolation. Finally, the physical parameters $\{G_{j,k}\}$ are uniquely recovered by inverting this linear system. The conditioning of $\mathbf{M}$ controls the final error amplification of the estimator; tuning $R'$ improves this conditioning and therefore reduces the amplification of statistical errors during inversion. The full procedure is detailed in Algorithm 2.

## 8. Experimental Results

### 8.1. Harmonic and Anharmonic Oscillators

To validate the first-quantized learning protocol described above, we numerically simulate the learning of a specific single-mode Hamiltonian. We start with the following Hamiltonian in first quantization:

$$\hat{H} = G_{2,0}\{\hat{x}^2\}_S + G_{0,2}\{\hat{p}^2\}_S + G_{4,0}\{\hat{x}^4\}_S, \qquad (22)$$

where $G_{2,0}$, $G_{0,2}$, and $G_{4,0}$ are the real physical coefficients to be learned. We specifically investigate two scenarios:

1. A Harmonic Oscillator: $\mathbf{G}_1 = [5.0, 2.5, 0]$.

2. An Anharmonic Oscillator: $\mathbf{G}_2 = [5.0, 2.5, 0.1]$.

Following the protocol in Section 7, we evaluate the estimation accuracy by computing the RMSE of the recovered coefficients with respect to the total evolution time $T$. As illustrated in Figure 4, for both harmonic and anharmonic cases, the estimation error converges to the Heisenberg limit scaling $\mathcal{O}(T^{-1})$ as the Trotter steps $L$ increase. This confirms that provided the D-RUT unitary $\hat{\mathcal{U}}(\kappa)$ is implemented with sufficient Trotter steps, the D-RUT protocol successfully captures $C(\beta)$ and recovers the physical parameters with Heisenberg-limited precision. Detailed theoretical analysis for the harmonic oscillator is shown in Appendix E.

We further investigate the protocol's resilience to SPAM errors, modeled as displacement noise: $\tilde{\beta}_j = \beta_j + \delta\beta_j$, with $\delta\beta_j \sim \mathcal{N}(0, \sigma_\beta^2)$. Figure 3 presents the estimation RMSE with respect to noise strength $\sigma_\beta$ for the anharmonic oscillator case, while the minimum of $|\beta_j|$ is set as 0.5. The results exhibit a linear scaling relationship, consistent with the theoretical bound $||\delta\mathbf{g}_{\text{SPAM}}||_2 \propto ||\delta\boldsymbol{\beta}||_2$ (see Appendix C) and thus confirms that our protocol remains robust under SPAM error.

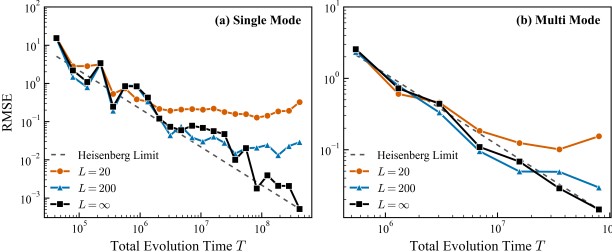

*Figure 5.* Heisenberg scaling verification for second-quantized Hamiltonian learning: (Left) Kerr oscillator and (Right) Bose-Hubbard dimer. Both cases exhibit the characteristic $T^{-1}$ scaling as the Trotter step density $L \to \infty$.

### 8.2. Kerr Oscillator and Bose-Hubbard Dimer

We numerically simulate the learning of two models:

1. **Single-Mode Kerr Oscillator:**

$$\hat{H}_{\text{Kerr}} = \omega \hat{n} + \chi \hat{n}^2 = \omega \hat{b}^\dagger \hat{b} + \chi(\hat{b}^\dagger \hat{b})^2. \qquad (23)$$

We aim to recover the frequency $g_{1,1} = \omega$ and Kerr nonlinearity $g_{2,2} = \chi$. Ground truth values are set to $\omega = 1.5, \chi = 0.5$.

2. **Bose-Hubbard Dimer with Cross-Kerr Interaction:**

$$\hat{H}_{\text{BH}} = \sum_{i=1}^{2}(\omega_i \hat{n}_i + U_i \hat{n}_i^2) + J(\hat{b}_1^\dagger \hat{b}_2 + \hat{b}_2^\dagger \hat{b}_1) + V \hat{n}_1 \hat{n}_2. \qquad (24)$$

The target coefficients are single-mode parameters $(\omega_i, U_i)$, linear coupling $J$, and nonlinear cross-Kerr coupling $V$. Ground truth values are set to $\boldsymbol{g} = [1.5, 0.5, 1.2, 0.4, 0.1, 0.05]$.

The numerical results are summarized in Figure 5. For both the Kerr oscillator and the Bose-Hubbard dimer, the estimation error follows the Heisenberg limit $\mathcal{O}(T^{-1})$ as the Trotter error is suppressed by increasing $L$.

## 9. Conclusion

We introduced D-RUT, an experimentally feasible active data acquisition and structured coefficient-recovery protocol for learning generic finite-order CV Hamiltonians at the Heisenberg limit. The framework uses designed physical probes to produce noisy polynomial responses and recovers Hamiltonian coefficients through stable reconstruction, with extensions to hierarchical multi-mode recovery and first-quantized Hamiltonian learning.

## Acknowledgements

DL acknowledges support from Beijing Municipal Science and Technology Commission and Zhongguancun Science Park Administrative Committee (No. 20251090054).

## Impact Statement

This work develops query-efficient tools for learning continuous-variable quantum dynamics, supporting calibration, verification, diagnosis, and benchmarking of quantum platforms used in quantum information processing and quantum machine learning. By improving Hamiltonian characterization under physical measurement constraints, the framework contributes to the scientific infrastructure needed for reliable quantum simulators, bosonic devices, and future QML experiments.

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

## A. Derivation of the Constant Term $C(\beta)$

The key to our protocol is the constant term $C(\beta)$, which is the eigenvalue of the effective Hamiltonian $\hat{\mathcal{H}}(\beta)$ for the vacuum state $|\text{vac}\rangle$. We now derive its analytical expression. After the displacement, a generic $g_{p,q}(\hat{b}^\dagger)^p \hat{b}^q$ in the original Hamiltonian $\hat{H}$ is first transformed as

$$\hat{D}^\dagger(\beta)\left(g_{p,q}(\hat{b}^\dagger)^p\hat{b}^q\right)\hat{D}(\beta) = g_{p,q}(\hat{b}^\dagger + \beta^*)^p(\hat{b} + \beta)^q \tag{25}$$

$$= g_{p,q}\left[\sum_{i=0}^{p}\binom{p}{i}(\hat{b}^\dagger)^i(\beta^*)^{p-i}\right]\left[\sum_{j=0}^{q}\binom{q}{j}\hat{b}^j\beta^{q-j}\right] \tag{26}$$

$$= g_{p,q}\sum_{i=0}^{p}\sum_{j=0}^{q}\binom{p}{i}\binom{q}{j}(\beta^*)^{p-i}\beta^{q-j}(\hat{b}^\dagger)^i\hat{b}^j. \tag{27}$$

Then we apply RUT to projects out all terms where $i \neq j$:

$$\mathbb{E}_{\theta\sim\mathcal{U}[0,2\pi]}\left[\mathbf{U}^\dagger(\theta)\hat{D}^\dagger(\beta)(g_{p,q}(\hat{b}^\dagger)^p\hat{b}^q)\hat{D}(\beta)\mathbf{U}(\theta)\right] = g_{p,q}\sum_{i=0}^{\min(p,q)}\binom{p}{i}\binom{q}{i}(\beta^*)^{p-i}\beta^{q-i}(\hat{b}^\dagger)^i\hat{b}^i. \tag{28}$$

To find the contribution to the total constant term $C(\beta)$, we select the term where $i = 0$:

$$C_{p,q}(\beta) = g_{p,q}\binom{p}{0}\binom{q}{0}(\beta^*)^p\beta^q(\hat{b}^\dagger)^0\hat{b}^0 = g_{p,q}(\beta^*)^p\beta^q. \tag{29}$$

Finally, summing over all terms in the original Hamiltonian gives the total constant term:

$$C(\beta) = \sum_{0<p+q\leq d} g_{p,q}(\beta^*)^p\beta^q. \tag{30}$$

This connects a measurable $C(\beta)$ and the target unknown coefficients $\{g_{p,q}\}$.

## B. Error Propagation Analysis

We assume that each measurement of $C(\beta_j)$ is independent with variance $\epsilon_C^2$. We now trace how this initial measurement statistical error propagates through the recovery process.

### B.1. Error Propagation in Radial Interpolation

For a fixed angle $\theta$, the recovery of the intermediate coefficient vector $\mathbf{g}_l(\theta) = [g_1(\theta),\ldots,g_d(\theta)]^T$ from the measurement vector $\mathbf{y} = [C(r_1,\theta),\ldots,C(r_{d+1},\theta)]^T$ is a linear system problem $\mathbf{y} \approx \mathbf{L}\tilde{\mathbf{g}}_l(\theta)$, where $\mathbf{L}$ is a Vandermonde matrix constructed from the Chebyshev radial nodes $\{r_\mu\}$, $\tilde{\mathbf{g}}_l$ is the real intermediate coefficients. To minimize $||\mathbf{L}\tilde{\mathbf{g}}_l(\theta) - \mathbf{y}||_2^2$, we utilize the least-squares method by defining $V = (\mathbf{L}\tilde{\mathbf{g}}_l(\theta) - \mathbf{y})^\dagger(\mathbf{L}\tilde{\mathbf{g}}_l(\theta) - \mathbf{y})$. We assume:

$$\frac{\partial V}{\partial\tilde{\mathbf{g}}_l^\dagger(\theta)} = \mathbf{L}^\dagger\mathbf{L}\tilde{\mathbf{g}}_l(\theta) - \mathbf{L}^\dagger\mathbf{y} = 0 \implies \tilde{\mathbf{g}}_l(\theta) = (\mathbf{L}^\dagger\mathbf{L})^{-1}\mathbf{L}^\dagger\mathbf{y} = \mathbf{L}^+\mathbf{y}, \tag{31}$$

where $\mathbf{L}^+$ is the pseudoinverse matrix of $\mathbf{L}$.

The estimation error $\delta\mathbf{g}_l = \mathbf{g}_l - \tilde{\mathbf{g}}_l$ is related to the measurement error $\delta\mathbf{y}$ with covariance $\text{Cov}(\delta\mathbf{y}) = \epsilon_C^2\mathbf{I}$ through $\mathbf{L}^+$. The covariance matrix of the estimated intermediate coefficients is thus given by:

$$\text{Cov}(\delta\mathbf{g}_l(\theta)) = \epsilon_C^2\mathbf{L}^+(\mathbf{L}^+)^\dagger = \epsilon_C^2(\mathbf{L}^\dagger\mathbf{L})^{-1}. \tag{32}$$

where we use the property that $\mathbf{L}^+(\mathbf{L}^+)^\dagger = ((\mathbf{L}^\dagger\mathbf{L})^{-1}\mathbf{L}^\dagger)(\mathbf{L}(\mathbf{L}^\dagger\mathbf{L})^{-1}) = (\mathbf{L}^\dagger\mathbf{L})^{-1}$. Thus the variance of a certain $g_l(\theta)$ is the corresponding $l$ th diagonal element:

$$\text{Var}(\delta g_l(\theta)) = \epsilon_C^2[(\mathbf{L}^\dagger\mathbf{L})^{-1}]_{ll}. \tag{33}$$

### B.2. Error Propagation in inverse Fourier Transform

For a fixed $l$, we solve for the target coefficient vector $\mathbf{g}_{p,l-p} = [g_{0,l}, g_{1,l-1}, \ldots, g_{l,0}]^T$ from the vector of intermediate values $\mathbf{g}_l = [g_l(\theta_0), \ldots, g_l(\theta_l)]^T$. This is another linear inversion, $\mathbf{g}_{p,l-p} \approx \mathbf{F}_l^{-1}\hat{\mathbf{g}}_l$, where $\mathbf{F}_l$ is the discrete Fourier transform matrix. The errors from the radial part propagate to the final coefficients as:

$$\mathrm{Cov}(\delta\mathbf{g}_{p,l-p}) = \mathbf{F}_l^{-1}\mathrm{Cov}(\delta\mathbf{g}_l)(\mathbf{F}_l^{-1})^\dagger. \tag{34}$$

We define the total Mean Squared Error (MSE) for $\{\mathbf{g}_l\}$ as $\epsilon_{g,l}^2$, which is the trace of Eq. 34. Since the $\mathbf{F}_l$ is unitary up to a factor. Thus, we have

$$\epsilon_{g,l}^2 = \mathrm{Tr}[\mathrm{Cov}(\delta\mathbf{g}_{p,l-p})] = \frac{1}{l+1}\mathrm{Tr}[\mathrm{Cov}(\delta\hat{\mathbf{g}}_l)] = \frac{1}{l+1}\sum_{u=0}^{l}\mathrm{Var}(\delta g_l(\theta_u)). \tag{35}$$

This result shows that the final estimation error is controlled by the RPE measurement precision, $\epsilon_C$, and the summation of $\frac{1}{\lambda_l}$, where $\lambda_l$ is the eigenvalues of the Gram matrix $\mathbf{G} = \mathbf{L}^\dagger\mathbf{L}$.

## C. Robustness under SPAM Errors

We now analyze the protocol's robustness against State Preparation and Measurement (SPAM) error. Specifically, we focus on the inaccurate implementation of the displacement parameter in practice. We model this error as $\tilde{\beta}_j = \beta_j + \delta\beta_j$, where $\tilde{\beta}_j$ is the real displacement with a small deviation, $\delta\beta_j$.

From the error propagation analysis, we can simply define an overall propagation matrix $\mathbf{K}$ consistent with the recovery strategy: Chebyshev interpolation followed by the angular inverse Fourier transform. Thus the target coefficients are given by $\mathbf{g} = \mathbf{K}^+\mathbf{y}$. Given that both statistical noise and SPAM errors are considered, the vector of actual measurement outcomes $\tilde{\mathbf{y}}$ is

$$\tilde{\mathbf{y}} = \mathbf{C}(\tilde{\boldsymbol{\beta}}) + \delta\mathbf{y}, \tag{36}$$

where $\mathbf{C}(\tilde{\boldsymbol{\beta}})$ is the vector of actual constant terms evaluated under displacements with deviation, and $\delta\mathbf{y}$ is the statistical noise from RPE. The real estimated coefficients are $\tilde{\mathbf{g}} = \mathbf{K}^+\tilde{\mathbf{y}}$ and the total error is therefore decomposed into the SPAM error part and the RPE statistical noise part:

$$\delta\mathbf{g}_{\text{total}} = \tilde{\mathbf{g}} - \mathbf{g} \tag{37}$$

$$= \mathbf{K}^+\left(\mathbf{C}(\tilde{\boldsymbol{\beta}}) + \delta\mathbf{y}\right) - \mathbf{g} \tag{38}$$

$$= \mathbf{K}^+\left([\mathbf{C}(\tilde{\boldsymbol{\beta}}) - \mathbf{C}(\boldsymbol{\beta})] + \mathbf{C}(\boldsymbol{\beta}) + \delta\mathbf{y}\right) - \mathbf{g} \tag{39}$$

$$= \underbrace{\left(\mathbf{K}^+\left(\mathbf{C}(\tilde{\boldsymbol{\beta}}) - \mathbf{C}(\boldsymbol{\beta})\right)\right)}_{\delta\mathbf{g}_{\text{SPAM}}} + \underbrace{\mathbf{K}^+\delta\mathbf{y}}_{\delta\mathbf{g}_{\text{RPE}}}, \tag{40}$$

where we use the fact that in an ideal case $\mathbf{g} = \mathbf{K}^+\mathbf{C}(\boldsymbol{\beta})$. Given that the statistical noise part has been analyzed in the previous section, we now focus on bounding the SPAM error term, $\delta\mathbf{g}_{\text{SPAM}}$. Taking the vector 2-norm, we find that

$$||\delta\mathbf{g}_{\text{SPAM}}||_2 = ||\mathbf{K}^+\left(\mathbf{C}(\tilde{\boldsymbol{\beta}}) - \mathbf{C}(\boldsymbol{\beta})\right)||_2$$

$$\leq ||\mathbf{K}^+||_2 \cdot ||\mathbf{C}(\tilde{\boldsymbol{\beta}}) - \mathbf{C}(\boldsymbol{\beta})||_2. \tag{41}$$

The norm of the pseudoinverse is given by the reciprocal of the smallest non-zero singular value of $\mathbf{K}$, $||\mathbf{K}^+||_2 = 1/\sigma_{\min}(\mathbf{K})$. And the second term can be bounded as

$$||\mathbf{C}(\tilde{\boldsymbol{\beta}}) - \mathbf{C}(\boldsymbol{\beta})||_2 \leq L_C||\delta\boldsymbol{\beta}||_2, \tag{42}$$

where $L_C$ is the Lipschitz constant. Combining these results yields the final upper bound of the SPAM error:

$$||\delta\mathbf{g}_{\text{SPAM}}||_2 \leq \frac{L_C}{\sigma_{\min}(\mathbf{K})}||\delta\boldsymbol{\beta}||_2. \tag{43}$$

This result demonstrates that the error in the final coefficients is linearly proportional to the magnitude of the displacement deviation, $||\delta\boldsymbol{\beta}||_2$, and the amplification factor depends on the condition number of $\mathbf{K}$ and the measurement which captured by $L_C$. Finally, we conclude that the protocol's robustness is controllable under careful selection of displacement $\{\beta_j\}$, which ensures a well-defined matrix $\mathbf{K}$.

### C.1. Bounding the Lipschitz Constant $L_C$

To complete the analysis, we provide a bound for the Lipschitz constant $L_C = \sup_{r,\theta} ||\nabla C(r,\theta)||_2$. We bound its radial and angular components of the gradient separately. In polar coordinates, $C(r,\theta)$ is given by $C(r,\theta) = \sum_{l=1}^{d} r^l g_l(\theta)$, thus the radial derivative is:

$$\frac{\partial C}{\partial r} = \sum_{l=1}^{d} l r^{l-1} g_l(\theta). \tag{44}$$

Given that $|g_l(\theta)| \leq \sum_{p+q=l} |g_{p,q}|$, if we assume the coefficients are bounded, $|g_{p,q}| \leq 1$, thus $|g_l(\theta)| \leq l+1$ and the magnitude of the radial derivative is bounded by:

$$\left|\frac{\partial C}{\partial r}\right| \leq \sum_{l=1}^{d} l r^{l-1}(l+1) \leq \sum_{l=1}^{d} l(l+1) r_{\max}^{l-1}. \tag{45}$$

If $|g_{p,q}| \geq 1$, thus we have $\left|\frac{\partial C}{\partial r}\right| \leq \sum_{l=1}^{d} l r_{\max}^{l-1}(\sum_{p+q=l} |g_{p,q}|)$. Similarly, for the angular component we have

$$\left|\frac{1}{r}\frac{\partial C}{\partial\theta}\right| \sim \sum_{l=1}^{d} \mathcal{O}(l^2) r_{\max}^{l-1}. \tag{46}$$

Combining these result provides an upper bound on $L_C$ that depends on the maximum order $d$, the maximum magnitude of displacement $r_{\max}$, and the summation of $\{|g_{p,q}|\}$.

## D. Comparative Analysis of Statistical Efficiency

We rigorously prove that our hierarchical D-RUT strategy yields a strictly lower worst-case estimation error compared to the simultaneous recovery strategy (as formulated in Appendix B.2 of (Möbus et al., 2025)).

As both protocols employ a "divide-and-conquer" approach to decompose a large-scale system into independent subsystems (clusters) of size $N \sim \mathcal{O}(1)$. The comparison below focuses on the learning efficiency within an independent $N$-mode subsystem. While the strategy in (Möbus et al., 2025) performs simultaneous parameter recovery on the $N$-dimensional hypercube, our D-RUT protocol further exploits the control of displacements ($\beta = 0$) to reduce the learning problem to a lower dimension $K = |S|$ (where $2 \leq |S| \leq N$ is the number of modes in the interaction term).

### D.1. Preliminaries

Let the measurement expectation value within a subsystem be a polynomial $P(\mathbf{x})$ of $N$ variables $\mathbf{x} = (x_1, \ldots, x_N)$ defined on a hypercube domain $\Omega = \prod_{\mu=1}^{N}[a_\mu, b_\mu]$, where $a_\mu > 0$ for all $\mu$.

We define the extrapolation ratio for the $\mu$-th mode as:

$$\rho_\mu := \frac{2|a_\mu|}{|b_\mu - a_\mu|}. \tag{47}$$

The condition $|b_\mu - a_\mu| > 2|a_\mu|$ in (Möbus et al., 2025) implies $0 < \rho_\mu < 1$. We generalize the error bound from Lemma D.1 of (Möbus et al., 2025) to the multivariate case and consider the recovery of the coefficient associated with the order $\mathbf{n} = (n_1, \ldots, n_N)$.

**Lemma D.1.** *Let $P(\mathbf{x})$ be a polynomial of $N$ variables ($N \in \mathbb{Z}^+$) with degree at most $d_\mu \leq d$ in each variable $x_\mu$. Assume $\tilde{P}(\mathbf{x})$ satisfy $|\tilde{P}(\mathbf{x}) - P(\mathbf{x})| \leq \epsilon$ for all $\mathbf{x} \in \prod_{\mu=1}^{N}[a_\mu, b_\mu]$. The error in the estimated coefficient $\tilde{p}_{\mathbf{n}} = \frac{1}{\mathbf{n}!}\partial^{\mathbf{n}}\tilde{P}(\mathbf{0})$ is strictly bounded by:*

$$|\tilde{p}_{\mathbf{n}} - p_{\mathbf{n}}| \leq \frac{\epsilon}{\mathbf{n}!} \prod_{\mu=1}^{N} \left( d_\mu(2d_\mu - 2)!! \left|\frac{2}{b_\mu - a_\mu}\right|^{n_\mu} \frac{1}{1 - \rho_\mu} \right). \tag{48}$$

*Proof.* Let $\delta P(\mathbf{x}) = \tilde{P}(\mathbf{x}) - P(\mathbf{x})$ be the error polynomial. The error in the coefficient is given by the mixed partial derivative:

$$\delta p_{\mathbf{n}} = \frac{1}{\mathbf{n}!} \partial^{\mathbf{n}} \delta P(\mathbf{0}). \tag{49}$$

We perform a multivariate Taylor expansion at sampling domain $\mathbf{a} = (a_1, \ldots, a_N)$:

$$\partial^{\mathbf{n}} \delta P(\mathbf{0}) = \sum_{\mathbf{k} \geq \mathbf{n}} \frac{1}{(\mathbf{k} - \mathbf{n})!} \partial^{\mathbf{k}} \delta P(\mathbf{a}) (-\mathbf{a})^{\mathbf{k} - \mathbf{n}}. \tag{50}$$

We bound the derivative at the boundary $|\partial^{\mathbf{k}} \delta P(\mathbf{a})|$ using the multivariate Markov Brothers' inequality:

$$|\partial^{\mathbf{k}} \delta P(\mathbf{a})| \leq \left( \prod_{\mu=1}^{N} \left| \frac{2}{b_\mu - a_\mu} \right|^{k_\mu} C_M(d_\mu, k_\mu) \right) \epsilon, \tag{51}$$

where

$$C_M(d, k) = \frac{d^2(d^2 - 1^2) \cdots (d^2 - (k-1)^2)}{(2k-1)!!} \leq d(2d-2)!!. \tag{52}$$

Substituting this back into the Taylor expansion and taking the absolute value:

$$|\delta p_{\mathbf{n}}| \leq \frac{1}{\mathbf{n}!} \sum_{\mathbf{k} \geq \mathbf{n}} \frac{1}{(\mathbf{k} - \mathbf{n})!} |\mathbf{a}|^{\mathbf{k} - \mathbf{n}} \left( \prod_{\mu=1}^{N} \left| \frac{2}{b_\mu - a_\mu} \right|^{k_\mu} C_M(d_\mu, k_\mu) \right) \epsilon. \tag{53}$$

We rearrange the summation and product to obtain:

$$|\delta p_{\mathbf{n}}| \leq \frac{\epsilon}{\mathbf{n}!} \prod_{\mu=1}^{N} \left[ \sum_{k_\mu = n_\mu}^{d_\mu} \frac{C_M(d_\mu, k_\mu)}{(k_\mu - n_\mu)!} \left| \frac{2}{b_\mu - a_\mu} \right|^{k_\mu} |a_\mu|^{k_\mu - n_\mu} \right] \tag{54}$$

$$\leq \frac{\epsilon}{\mathbf{n}!} \prod_{\mu=1}^{N} \left[ d_\mu(2d_\mu - 2)!! \left| \frac{2}{b_\mu - a_\mu} \right|^{n_\mu} \sum_{k_\mu = n_\mu}^{d_\mu} \frac{1}{(k_\mu - n_\mu)!} \left( \frac{2|a_\mu|}{b_\mu - a_\mu} \right)^{k_\mu - n_\mu} \right] \tag{55}$$

$$= \frac{\epsilon}{\mathbf{n}!} \prod_{\mu=1}^{N} \left[ d_\mu(2d_\mu - 2)!! \left| \frac{2}{b_\mu - a_\mu} \right|^{n_\mu} \sum_{j=0}^{d_\mu - n_\mu} \frac{1}{j!} \rho_\mu^j \right] \tag{56}$$

$$\leq \frac{\epsilon}{\mathbf{n}!} \prod_{\mu=1}^{N} \left[ d_\mu(2d_\mu - 2)!! \left| \frac{2}{b_\mu - a_\mu} \right|^{n_\mu} \frac{1}{1 - \rho_\mu} \right]. \tag{57}$$

Here we let $j = k_\mu - n_\mu$ and use the property $\frac{1}{j!} \leq 1$ for integer $j \geq 0$, thus we bound the partial sum by:

$$\sum_{j=0}^{d_\mu - n_\mu} \frac{1}{j!} \rho_\mu^j \leq \sum_{j=0}^{\infty} \rho_\mu^j = \frac{1}{1 - \rho_\mu}. \tag{58}$$

$\square$

## D.2. Error Analysis

In the simultaneous strategy, all coefficients are recovered on the full $N$-dimensional hypercube $\Omega_{\text{sim}} = \prod_{\mu=1}^{N} [a_\mu, b_\mu]$. Consider a coupling coefficient $c_{\mathbf{p}_S, \mathbf{q}_S}^{(S)}$ associated with a cluster of modes $S$. This coefficient corresponds to a multi-index $\mathbf{n}$ where $n_\mu = p_\mu + q_\mu > 0$ for $\mu \in S$ (as defined in Eq. 1) and $n_\nu = 0$ for $\nu \notin S$.

Applying Lemma D.1 with dimension $M = N$, the error bound for the simultaneous strategy is:

$$|\delta c_{\text{sim}}^{(S)}| \leq \frac{\epsilon}{\mathbf{n}!} \left( \prod_{\mu \in S} C_\mu \frac{1}{1 - \rho_\mu} \right) \cdot \left( \prod_{\nu \notin S} C_\nu \frac{1}{1 - \rho_\nu} \right), \tag{59}$$

where the constant $\mathcal{C}_\mu$ for active modes ($\mu \in S$) and $\mathcal{C}_\nu$ for inactive modes ($\nu \notin S$) are:

$$\mathcal{C}_\mu = d_\mu(2d_\mu - 2)!! \left| \frac{2}{b_\mu - a_\mu} \right|^{n_\mu}, \tag{60}$$

$$\mathcal{C}_\nu = d_\nu(2d_\nu - 2)!! \left| \frac{2}{b_\nu - a_\nu} \right|^0 = d_\nu(2d_\nu - 2)!!. \tag{61}$$

Our hierarchical recovery strategy utilizes the physical capability to set displacement $\beta_j = 0$. This allows us to perform coefficient recovery on strictly lower dimension. To learn all the single and coupling coefficients of cluster $S$, we set $\beta_\nu = 0$ for all $\nu \notin S$. The extrapolation collapses to the $|S|$-dimensional domain $\Omega_S = \prod_{\mu \in S}[a_\mu, b_\mu]$.

Applying Lemma D.1 with dimension $M = |S|$, we similarly obtain:

$$|\delta c_{\text{hie}}^{(S)}| \le \frac{\epsilon}{\mathbf{n}!} \prod_{\mu \in S} \mathcal{C}_\mu \frac{1}{1 - \rho_\mu} = \frac{\epsilon}{\mathbf{n}!} \prod_{\mu \in S} \left( d_\mu(2d_\mu - 2)!! \left| \frac{2}{b_\mu - a_\mu} \right|^{n_\mu} \frac{1}{1 - \rho_\mu} \right). \tag{62}$$

We compare the upper bounds derived in Eq. (59) and Eq. (62):

$$\frac{|\delta c_{\text{sim}}^{(S)}|}{|\delta c_{\text{hie}}^{(S)}|} = \prod_{\nu \notin S} \mathcal{C}_\nu \frac{1}{1 - \rho_\nu} > 1. \tag{63}$$

Since $\mathcal{C}_\nu \ge 1$ and $\frac{1}{1-\rho_\nu} > 1$ for all $\nu$. This inequality holds strictly for any $N > |S|$ and the ratio grows exponentially as:

$$\frac{|\delta c_{\text{sim}}^{(S)}|}{|\delta c_{\text{hie}}^{(S)}|} \sim \mathcal{O}\left( \left( \frac{1}{1 - \rho_\nu} \right)^{N - |S|} \right). \tag{64}$$

Note that we assume a constant and equivalent error bound $\epsilon$ for both strategies to focus on the error amplification due to extrapolation. However, achieving this bound typically requires significantly more resources in the simultaneous strategy, which further strengthens the statistical efficiency for the hierarchical recovery strategy.

## E. Numerical scheme: Learning of a specific Harmonic Oscillator

We provide a numerical scheme to validate the first quantization learning protocol. We start with the following Hamiltonian in the first quantization:

$$\hat{H} = G_{2,0}\{\hat{x}^2\}_S + G_{0,2}\{\hat{p}^2\}_S = G_{2,0}\hat{x}^2 + G_{0,2}\hat{p}^2, \tag{65}$$

where $G_{2,0}$ and $G_{0,2}$ are arbitrary real coefficients to be learned.

Substituting Eq. (20) into the Hamiltonian $\hat{H}$:

$$\hat{H} = \frac{\hbar}{2} \left[ \frac{G_{2,0}}{m_0\omega_0} e^{2R'} (\hat{B}' + \hat{B}'^\dagger)^2 - G_{0,2}m_0\omega_0 e^{-2R'} (\hat{B}' - \hat{B}'^\dagger)^2 \right] \tag{66}$$

$$= g'_{1,1}\hat{B}'^\dagger \hat{B}' + g'_{2,0}(\hat{B}^\dagger)^2 + g'_{0,2}\hat{B}'^2 + g'_{0,0}. \tag{67}$$

where the coefficients $g'_{p,q}$ now explicitly depend on both the reference frame and $R'$:

$$g'_{2,0} = g'_{0,2} = \frac{\hbar}{2} \left( \frac{G_{2,0}}{m_0\omega_0} e^{2R'} - G_{0,2}m_0\omega_0 e^{-2R'} \right), \tag{68}$$

$$g'_{1,1} = \frac{\hbar}{2} \left( \frac{2G_{2,0}}{m_0\omega_0} e^{2R'} + 2G_{0,2}m_0\omega_0 e^{-2R'} \right), \tag{69}$$

$$g'_{0,0} = \frac{\hbar}{2} \left( \frac{G_{2,0}}{m_0\omega_0} e^{2R'} + G_{0,2}m_0\omega_0 e^{-2R'} \right). \tag{70}$$

The measurable $C(\beta)$ is derived as:

$$C(\beta) = g'_{2,0}(\beta^*)^2 + g'_{0,2}\beta^2 + g'_{1,1}|\beta|^2. \tag{71}$$

By solving the linear system $\mathbf{g}' = \mathbf{MG}$, we obtain:

$$G_{2,0} = \frac{m_0\omega_0 e^{-2R'}}{\hbar}\left(\frac{1}{2}g'_{1,1} + g'_{2,0}\right), \tag{72}$$

$$G_{0,2} = \frac{e^{2R'}}{\hbar m_0\omega_0}\left(\frac{1}{2}g'_{1,1} - g'_{2,0}\right). \tag{73}$$

Provided that the mismatch between the reference frame $(m_0, \omega_0)$ and the physical parameters $(m, \omega)$ is bounded, setting the experimental squeezing parameter $R' = 0$ is generally sufficient for robust recovery. However, if the reference frame differs significantly from the true physical system, the condition number of $\mathbf{M}$ may degrade, amplifying statistical noise.

To mitigate this, we utilize the non-diagonal coefficient $g'_{2,0}$ as a signal function. Considering the physical definitions where $G_{2,0} = \frac{1}{2}m\omega^2$ and $G_{0,2} = \frac{1}{2m}$, the coefficient $g'_{2,0}$ is explicitly given by:

$$g'_{2,0} = \frac{\hbar}{4}\left(\frac{m\omega^2}{m_0\omega_0}e^{2R'} - \frac{m_0\omega_0}{m}e^{-2R'}\right). \tag{74}$$

A non-zero value of $g'_{2,0}$ indicates a deviation between the experimental basis and the system's basis. This suggests a straightforward optimization strategy: by iteratively tuning $R'$ to minimize the magnitude $|g'_{2,0}|$, we effectively reduce the relative mismatch and improve the condition number of $\mathbf{M}$, ensuring numerically stable recovery of the physical coefficients.

