# OpenReview forum: "Continuous Variable Hamiltonian Learning at Heisenberg Limit via Displacement-Random Unitary Transformation"
_ICML.cc/2026/Conference — ICML 2026 regular_

### Official Review · Reviewer_AEXa · 2026-02-28

**Soundness:** 3
**Presentation:** 1
**Significance:** 1
**Originality:** 1
**Overall Recommendation:** 1
**Confidence:** 5

**Summary:**

This work proposes a novel framework of D-RUT which learns the coefficients of a continuous-variable Hamiltonian while meeting the Heisenberg error scaling in both single and multi mode nonlinear systems. It also expands prior Hamiltonian learning domain in to the first quantization and also improves estimation error bound compared to prior works with its hierarchical recovery system, while remaining robust against state preparation and measurement error.

**Compliance With Llm Reviewing Policy:**

Affirmed.

**Final Justification:**

# Final Recommendation: Strong Rejection (1)

After carefully reading the manuscript and the authors' rebuttals, I am very confident that D-RUT is not a machine learning paper. It is a quantum metrology / Hamiltonian estimation paper, and if it were to be accepted as an ML paper, any metrology paper should be accepted as an ML paper as long as it is statistically estimating the ground-truth quantity with controlled measurements; which are all of them. Reviewer fjKS and 37Rw also seems to be unconvinced on the fit of this paper to ICML, and I am now confident that it does not fit to ICML indeed.

Primary reason why D-RUT is not a machine learning method is because the query policy is predetermined and is not adaptive to the previous stage of learning. Please refer to my rebuttal acknowledgement on my reasonings. I think the manuscript is well-written and technically solid. However, I am very against accepting this paper to publish on ICML, an ML venue, as I believe this work is best defined as a quantum metrology / estimation paper as there is no learning-theoretic component in D-RUT.

# On the authors' reply rebuttals

I have carefully read through the authors' reply rebuttals, and I am not convinced to the authors' claims and still remain very confident that D-RUT is best defined as a metrology / system identification work not a learning-theoretic paper. The authors retract their claim on the active learning, and that makes me feel that the authors were not rigorous and careful on their first rebuttal, and gives me the impression that the machine learning framing the authors are trying is very hasty and unprepared, ad hoc to publish it on a ML venue. That being said, the reply rebuttal's main framing as a statistical learning contribution is also mostly brittle and erroneous with the reasons below:

1. The labeling of model / parameters/ query / outcome / loss the authors suggest still remain to be **just labels**. **Almost any inverse problem can be written in that framing**: spectroscopy, tomography, curve fitting, system identification, all admit a likelihood, parameters, and a least-squared objective; **that alone does not make them statistical learning contributions in ML standards**. D-RUT's technical gravity is still centered on **optimal experimental design / SPAM (Hardware error) robustness / and quantum algorithmic reconstructions** from **predetermined displacement grid** (the Chebyshev radii and sampling angles, etc.).

2. Authors claim D-RUT also solved a prediction / testing-error problem in the sense of Dutt et al. (https://journals.aps.org/prresearch/abstract/10.1103/PhysRevResearch.5.033060), but it is not true. Dutt et al. **explicitly define** a **query space, testing distribution**, and separate **"model inference"** from **"prediction against a testing distribution"**. That is the statistical learning contribution in ML sense. In contrast, D-RUT does not present testing distribution nor a theorem on the expected test error under an arbitrary testing distribution. What D-RUT proves is recovery/stability on a fixed desinged domain via interpolation: the authors' claim that it gives **universal convergence** over the bounded betas, **does NOT suffice to be universal convergence from empirical to population risk in statistical learning notion**. That is why I believe D-RUT is an estimation technique, not a statistical learning technique; **D-RUT processes the measurements into a desired formulation, but it is NOT about learning a proxy model that aims to minimize population risk over an unknown distribution**.

3. The authors' comparison to classical shadows is not helpful. Classical shadows indeed use predetermined randomized measurements, but the key output is a **reusable classical representation** from which the user can **reuse on many different target properties**, and those target properties can even be chosen **after the measurements are completed** (https://www.nature.com/articles/s41567-020-0932-7). THAT would suffice a genuine prediction / learning problem, I would fully agree. Another nice example would be Gong et al. (https://proceedings.mlr.press/v202/gong23a.html), where they explicitly formulates a learning-theoretic task of estimating **unknown outcome distributions of unknown measurements in total variation distance**. I am not suggesting D-RUT MUST follow the formulation of those references, I am trying to make some examples of related application works that truly formulate learning-theoretic tasks, I hope the authors would consider my point. In contrast, D-RUT estimates a **fixed finite coefficient vector** in a **known Hamiltonian ansatz** using a **fixed experimental design** and **analytic inversion**. I am also not suggesting that any one of those components disqualify D-RUT as a statistical learning contribution, but that the **formulation in a whole** is not a statistical learning contribution. I hope the authors would consider my point.

4. The **"active data acquisition"** is more naturally pointing to **optimal experimental design / system identification / metrology** contribution than to statistical learning, without an adequate learning-theoretic formulation as I have explained above. From what I have found, there indeed is a Hamiltonian learning paper that focuses heavily on the system identification and estimation as much as D-RUT, but which actually sits nicely in learning-theoretic contribution as well. Granade et al. (https://arxiv.org/abs/1207.1655) explicitly combines **sequential Monte Carlo** and **Bayesian experimental design** to perform online Hamiltonian learning **during** data collection. That kind of concrete formulation would be a statistical learning contribution indeed.


### Conclusion

The reply rebuttal is valid only at the level of **"D-RUT is a statistical estimation with a probabilistic model"**, which does NOT make it a statistical learning contribution. The main reason is **NOT** because D-RUT is **"too analytical"**, as the authors seem to believe. The main reason is that D-RUT is formulated on a **physics-designed inverse problem**, making **probabilistic estimation** with **predetermined measurements protocol** and **predetermined recovery process**. Processing the measurement data into a desired format is what any metrology and inverse problem does. I regretfully maintain my assessment prior to the reply rebuttals and recommend **strong rejection** as the paper does not fit to ICML.

**Key Questions For Authors:**

1. What is the machine learning part of the D-RUT protocol? What is the training objective? What is the optimization problem here?
2. I notice that all prior works on Hamiltonian learning are from physics/quantum venues. Although the name contains the word "learning", what is the machine learning component in a generic Hamiltonian learning methods?
3. Could you please clarify the key intuition, ideas, novelty, and impact of this work in machine learning perspective? What is new about D-RUT as a machine learning method, and why is it impactful?
4. Is the number-conserving effective operator an ansatz? Or is it a measurement operator? Is it known/designed, or is it also an unknown black box?
5. What does the "projecting/subjecting" Hamiltonian to an operator mean in ML perspective? Robust Phase Estimation and hierarchical coefficient recoveries appear to be purely analytical, is the projecting to the operator part the learning component of D-RUT?

Overall, I would very much appreciate it if the authors would kindly explain the D-RUT protocol in the lens of machine learning, for instance, what is the training objective here and what is the optimization method and what is its contribution and impact to machine learning. I am very willing to raise the ratings to an acceptance level if the authors would help me see the ML contribution and impact of this work that I might have missed, and they would be willing to revise the manuscript thoroughly to make it "ML-friendly".

**Limitations:**

Could the authors please specify the potential impacts of this work? Authors wrote that they do not feel the necessity to specify them but I cannot imagine them myself based on the current manuscript.

**Strengths And Weaknesses:**

Strength
- D-RUT expands the Hamiltonian learning domain into first quantization
- D-RUT is more accurate than prior methods while still remaining robust
- Work proves the error bound and scaling guarantee, fairly theoretically complete.

Weakness
- Biggest problem of this work is that I cannot see any ML side contribution. Actually, I cannot understand if there is any ML component at all in the D-RUT protocol. From what I am understanding, all process in D-RUT protocol look analytical: project target Hamiltonian to an operator, measure, and extract encoded coefficients from the measurements with varying displacement by Chebyshev interpolation and DFT, I cannot see learning component anywhere. I am not even sure if this work can be categorized into a machine learning paper from what I am reading.
- Also, another major problem of this manuscript is that it is very opaque for a generic ML audience with all the quantum jargons and maths. For instance, what meaning does the "projecting/subjecting" target Hamiltonian to a number-conserving operator carry in ML sense? Is the operator ansatz? Is it a measurement? Is it a regression? What is the meaning of it in terms of machine learning? Terms such as "trotterization" are used without any explanation, and I strongly doubt if any ML audience can read this manuscript and clearly understand the key intuitions, ideas, steps, and impacts of this work. If this manuscript is to be accepted to a ML/AI venue, I strongly recommend the authors to revise the draft into languages and presentation friendly for a generic ML audience.
- Coming from the first and second problems, current manuscript does not clearly reveal what is its contribution and impact to machine learning, and yet the authors have not specified any impact statement which makes it harder to perceive the novelty and significance of this work.

In general, I strongly feel that this paper does not belong to a ML/AI venue. This work's impact and significance in the field of Hamiltonian Learning, I cannot presume of that, and if their assessment is true, this work might be very welcomed in a quantum/physics venue. However, at least based on the current manuscript, and regardless of the true value of this work for physics community, I cannot find any ML-side contribution in this work. I want to make it clear that my ratings are NOT for the physics/quantum-side quality of this work. For instance, significance/originality ratings are not about the physics/quantum aspect of this work, but just that I cannot see any contribution and novelty in the D-RUT protocol in the ML side.

---

> ### Author Rebuttal · Authors · 2026-03-31
>
> **Core question: What is learning here?**
> Hamiltonian learning is statistical learning under data constraints.
> Below is a terminology mapping:
> | ML Concept | Hamiltonian Learning (This work) |
> | :--- | :--- |
> | **Model / Hypothesis** | Parametric Hamiltonian (H(θ)) |
> | **Parameters** | Unknown coefficients (θ*) |
> | **Data** | Measurement outcomes |
> | **Query / Input design** | Displacement parameters (β) |
> | **Loss** | Estimation error |
> | **Training algorithm** | D-RUT (active measurement + reconstruction) |
>
>
> **Analogy:** Mirrors linear regression with optimal inputs: choosing β (active learning), observing noisy y, and reconstructing parameters via structured inversion (Chebyshev+Discrete Fourier Transformation(DFT)). D-RUT is an active learning algorithm with optimal queries and a closed-form estimator.
>
> Our setting highlights two Learning Theory contributions vs. classical ML:
> 1. **Joint acquisition optimization:** Quantum measurements are constrained. Designing β and number-preserving operations yields physical feature construction, mirroring optimal experimental design minimizing worst-case error.
> 2. **Analytical learning via structured models:** Hilbert space structure provides orthogonal bases/invertibility, enabling closed-form estimators with provable guarantees, avoiding the use of NNs/gradient descent. This aligns with classical spectral methods, compressed sensing, etc, and extends to quantum-constrained data generation.
>
> **ML Significance:** Efficient physical system learning is a core Quantum Machine Learning (QML) problem. Our work affirmatively answers: Can learning algorithms achieve provable quantum sample complexity advantages? and achieves the Heisenberg limit (T ~ O(1/ε) vs. classical T ~ O(1/ε^2)), establishing strict quantum advantage in statistical learning efficiency.
>
> **Q1:**
> - Target: Unknown coefficients g (ground-truth parameters).
> - Training Data: Measurement outcomes from physical queries.
> - Objective (Loss): Minimize estimation error (RMSE) under query budget (total time T).
> - Optimization: As gradient descent on exponential quantum spaces fails(barren plateaus), D-RUT uses analytical optimization. Active query design (β, RUT) transforms non-linear, infinite-dimensional optimization into tractable linear inversion, analogous to SVMs/Kernel Ridge Regression using convex optimization.
>
> **Q2:** Hamiltonian learning is essentially statistical learning in quantum information. Top venues increasingly feature analytical quantum learning: *Learning the Complexity of Weakly Noisy Quantum States* (ICLR 2025); *Performance Analysis of Quantum Machine Learning Classifiers* (NeurIPS 2025); *Understanding Generalization in QML with Margins* (ICML 2025); *Quantum machine learning advantages* (ICLR 2025).
>
> **Q3:** Four ML/QML contributions:
> 1. **Optimal Active Learning:** Selecting displacement at Chebyshev nodes minimizes worst-case error(optimal experimental design).
> 2. **Sample Complexity Advantage:** Provably achieving Heisenberg limit T ~ O(1/ε) shows strict efficiency advantages over classical T ~ O(1/ε^2).
> 3. **High-dimensional Inference:** Decomposing multi-mode learning mirrors exploiting sparsity/graphical models to beat the curse of dimensionality.
> 4. **Impact:** Characterizing Continuous Variable systems is prerequisite for QNNs. D-RUT generalizes to first-quantization and is experiment-friendly for near-term hardware, surpassing prior SOTA.
>
> **Q4:** In ML terms, its a derived **Feature Extraction/Dimensionality Reduction** mapping—neither an ansatz nor final measurement. Specific pulses (D-RUT) filter non-linear noise, collapsing the system into a clean, low-dimensional subspace (scalar C(β) ) for parameter decoding.
>
> **Q5:**
> Physically, quantum observation is a "projection." The learning component is the *entire* D-RUT protocol:
> * **Hierarchical Recovery -> Block-Coordinate Descent:** Resolving parameters by clusters improves statistical efficiency.
> * **Applying β -> Active Feature Selection:** Choosing specific inputs to probe the system.
> * **Applying RUT -> Pooling/Latent Representation Extraction:** Averaging redundant dimensions to project data into a linear, learnable latent subspace.
> * **RPE -> Noisy Label Collection:** Quantum uncertainty prevents full-state access, RPE is the robust statistical procedure to collect noisy oracle labels.
>
> **Planned Revisions:** We are adding a "Machine Learning Perspective" section framing our task as statistical parameter estimation via active queries, mapping quantum jargon to ML analogues (e.g., Trotterization ~ numerical time discretization). The expanded Impact Statement articulates how our Hamiltonian learning protocol provides critical hardware characterization infrastructure, essential for scaling quantum computing and robust quantum-enhanced ML.
>
> We thank the reviewer for the detailed feedback, hope these clarifications address your concerns.  We would truly appreciate it if the reviewer would consider raising the score.

---

> > ### Author Rebuttal · Reviewer_AEXa · 2026-04-02
> >
> > Thank you for the rebuttals and the revision plans reflecting my feedback. However, I am afraid I remain unconvinced that D-RUT is an active statistical learning algorithm. D-RUT is more suited to be categorized as a quantum metrology / system identification algorithm, and I remain to believe it is not a good fit to ICML, but belongs to physics venues. Below are the reasons why:
> >
> > # Why D-RUT is not active learning in ML sense
> >
> > Main reason D-RUT does not suffice to be an active learning algorithm in ML sense is because the **query policy is not learned from past observations, and predetermined before the data collection**. In standard active learning, the learner asks a query, updates from the answer, and then **uses that updated knowledge to decide what to query next** (https://minds.wisconsin.edu/handle/1793/60660). For example, please refer to Dutt et al. 2023 (https://journals.aps.org/prresearch/abstract/10.1103/PhysRevResearch.5.033060), which is an actual active Hamiltonian learning. In Dutt et al. 2023,  the learner is explicitly given an initial dataset, forms a current estimate of the Hamiltonian, and then **uses that current estimate to decide which queries to make next**. The method batches the queries **according to the current state** of learning, hence the next query distribution depends on the current estimate: this is an active learning.
> >
> > By contrast, D-RUT dos not have that outer adaptive loop, its probe schedule is essentially **pre-determined and fixed** during the entire process. in Algorithm 1 of D-RUT, Chebyshev radii and sampling angles are fixed **before** data collection, and the protocol runs through that **predetermined** set of $\beta$ values, and then goes another **fixed** reconstruction process. The only somewhat adaptive part is the RPE subroutine, but that is only subroutine level; overall query design is not adaptive at all. Therefore, D-RUT is **NOT** an active learning algorithm in any ML standard.
> >
> > # Why D-RUT is not statistical learning in ML sense
> >
> > Main reason why D-RUT does not suffice as a statistical learning in ML sense is the absence of **generalization-based learning formulation** over an unknown data distribution. Instead, it is better described as a **designed parameter-estimation procedure** under noisy quantum measurements. Statistical learning in ML is usually defined by a hypothesis class, an instance space, a loss/objective, and an unknown distribution over samples. The learner receives a finite sample, outputs a hypothesis, and success is judged by population risk / generalization to unseen data from that underlying true distribution (https://www.jmlr.org/papers/volume11/shalev-shwartz10a/shalev-shwartz10a.pdf). Dutt et al. 2023 also suffice to be a statistical learning, as it defines a query space, outcome space, training examples, a loss, and distinguishes model inference from prediction against a testing distribution.
> >
> > By contrast,
> > 1. D-RUT does **not** posit an unknown underlying distribution over samples nor generalization under that distribution. Instead, it performs **controlled experiments** at **designer-chosen** probe settings.
> > 2. D-RUT does **not** learn a predictor nor hypothesis whose population risk on unseen examples is the main object of study. What it aims for is on the **parameter-estimation error** versus **physical resources**.
> > 3. D-RUT's data acquisition and reconstruction map are largely **predetermined**, not learned from empirical data in a generalization-theoretic framework.
> >
> > Therefore, D-RUT is best described as a **statistical estimation method**, but **NOT** as a statistical learning method in any ML standard.
> >
> > # Overall Assessment
> >
> > In a broad sense, I agree that D-RUT may be assessed as **ML-adjacent**. However, the center of its technical gravity is much fitted to be of interest of **Hamiltonian estimation, quantum metrology, and system identification**, not to ML methodology nor ML theory, nor ML application. The acclaimed ML contribution in authors' rebuttal are invalid, since:
> > 1. D-RUT is **NOT** active learning in standard ML sense
> > 2. Physical resource scaling is **NOT** ML-theoretic contribution.
> > 3. This is **NOT** an ML contribution, but rather a **physics-specific coefficient-isolation procedure**, as the decomposition is manually imposed by controllable displacements and the recovery is **NOT** statistically learned.
> > 4. Impact on QNNs / near-term HW is application relevance, **NOT** an ML contribution.
> >
> > Overall, the methodology and contributions are best fit on **quantum Hamiltonian estimation, experimental design**, and **metrology**. D-RUT does not substantiate them as contributions to active / statistical learning, or ML methodology in the standard sense. I regretfully maintain my original rating, I genuinely think D-RUT is an impactful quantum estimation/metrology work; however, as an ICML reviewer, I am afraid it does not fit to a ML venue.

---

> > > ### Author Response · Authors · 2026-04-08
> > >
> > > ### 1. Formulating D-RUT Hamiltonian Learning as Statistical Learning
> > > We thank the reviewer for the follow-up. We note that **D-RUT is not a simple estimation theory.** Our learning object is Hamiltonian, a high-dimensional object parameterized by multiple coefficients. We map D-RUT to the definitions and equations in the same learning framework of Dutt et al.:
> > >
> > > * **Definitions:** Model: Unknown Hamiltonian; Parameters θ: Hamiltonian coefficients;  Query: x = (β, κ) (displacement β, evolution time κ); Outcome: ancilla measurement outcome. Query distribution: Joint space of β and κ.
> > >
> > > * **Conditional Probability (Eq. 1):** Our Eq. 11 defines the probabilistic generative model. Given query x = (β, κ), outcome |0> probability is P_0^Re = 1/2(1 + cos(κ C(β; θ))), where C(β; θ) encodes unknown parameters θ.
> > > * **Loss Function (Eq. 2):** D-RUT parameter reconstruction minimizes least-squares loss: V(θ) = ||Lθ - y_measured||^2_2 (Appendix B, Eq. 31, L:Vandermonde matrix; y_measured: noisy phases).
> > > * **Model Inference Objective (Eq. 3):** Our primary metric is RMSE in Hamiltonian coefficients as in **Problem II.1 (Model Inference)**.
> > > * **Estimation Procedure (Eq. 4):** Dutt uses stochastic gradient descent. D-RUT provides a closed-form analytical estimator.
> > > **Testing Error / Prediction (Eq. 5):*D-RUT solves the prediction task against testing error (Problem II.2). Given an arbitrary, unseen testing distribution p_test(β, κ) (|β| ≤ R_max), by learning from queried x, D-RUT solves parameters θ̂ via Chebyshev interpolation to guarantee uniform convergence, thus analytically bounds the expected prediction error (population risk) sampled from {|β|≤R_max |Ĉ(β) - C(β)|}.
> > >
> > > **Takeaway:** Statistical learning defines the problem, for which active learning is one way to solve it, while analytical inversion with “active” data acquisition (D-RUT)  is another solution. **Crucially, D-RUT is not merely a physical experimental tool; “active” data acquisition, and structured inversion together form a complete ML estimator.**
> > >
> > > QML community consensus holds that Hamiltonian learning fundamentally belongs to learning theory, regardless of adaptive/non-adaptive solvers. Classical Shadows(widely published at ICML/NeurIPS) uses a completely predetermined, non-adaptive random measurement. Similarly, D-RUT's non-adaptive exact parameter recovery under quantum constraints should also belong to ML contribution.
> > >
> > > ### 2. "Active" Data Acquisition vs. Active Learning
> > > We acknowledge our use of "active learning" was inaccurate, lacking the adaptive query loop required. **We retract this claim.**
> > >
> > > However, D-RUT utilizes "active" data acquisition, where data and measurements are not given passively but selected in advance. Selecting query points *before* data collection to minimize worst-case estimation error. The non-adaptive **Classical Shadows** framework shares same structural features the reviewer disqualifies in D-RUT:
> > >  * *Adaptive query loop?* No (predetermined random measurements).
> > >  * *Unknown distribution over samples?* No (controlled experiments with designed random unitaries).
> > >  * *Learn a predictor with population risk?* No (estimates expectation values tr(Oρ)).
> > >  * *Query design learned from data?* No (measurement ensemble fixed a priori).
> > >
> > > Classical Shadows with variants has been widely accepted as quantum *learning* and published at ML conferences (arXiv:2209.03007, 2305.13362, 2405.18489). D-RUT, solving a challenging Hamiltonian learning problem like quantum state learning,  with provably optimal query complexity, should follow in the same category of QML.
> > >
> > > ### 3. Responses to overall assessment
> > >
> > > > **Point 1:**
> > >
> > > We addressed these by mapping D-RUT to statistical learning, defining generalization error, noting D-RUT employs "active" data acquisition and highlighting alignment with established non-adaptive QML protocols like Classical Shadows.
> > >
> > > > **Point 2-4:**
> > >
> > > We respectfully disagree, since sampling complexity translates to physical resource cost. The fact that samples are quantum measurements rather than classical i.i.d. draws doesn’t change its learning-theoretic nature, aligns with prior work like Classical Shadows. We also disagree with the logic regarding manually imposed structures, as it would negate ML contributions like compressed sensing and classical shadows. We agree to list application relevance as motivation only, and our core contribution is developing a provably efficient Hamiltonian learning algorithm with sample complexity achieving Heisenberg-limited scaling.
> > >
> > > ***
> > >
> > > We sincerely thank the reviewer for discussions that significantly improve our work. We will include the explicit statistical learning mapping in the updated version. Acknowledging the conceptual gap between domains, we aim to bridge it and advance QML theory with demonstrated theoretical and experimental impact. We would greatly appreciate any further suggestions, and hope that these clarifications may help increase the reviewer's overall evaluation.

---

### Official Review · Reviewer_fjKS · 2026-03-08

**Soundness:** 4
**Presentation:** 4
**Significance:** 3
**Originality:** 3
**Overall Recommendation:** 5
**Confidence:** 4

**Summary:**

As a preliminary note, my primary expertise lies in general quantum information and DV rather than in the CV domain. I have evaluated this manuscript based on the fundamental principles of Hamiltonian learning and quantum parameter estimation.

This paper introduces the Displacement-Random Unitary Transformation (D-RUT), an elegant protocol designed to learn the coefficients of generic multi-mode bosonic Hamiltonians of arbitrary finite order. By combining displacement operations with random phase rotations, the authors ingeniously project unbounded, off-diagonal operators into a particle-number-conserving diagonal subspace. Coupled with Robust Phase Estimation (RPE) and classical Chebyshev interpolation, the protocol theoretically achieves Heisenberg-limited scaling for parameter extraction.

**Compliance With Llm Reviewing Policy:**

Affirmed.

**Final Justification:**

I have carefully read the authors' rebuttal and the other reviews. I am upgrading my score to an Accept (5) and am willing to champion this paper for acceptance.

I completely understand the strong reservations raised by Reviewer AExa. From the standpoint of traditional ML optimization, the methodology undeniably lacks the conventional algorithmic "flavor". However, looking through the lens of PAC Learning, the authors have successfully and rigorously mapped their analytical inversion protocol into a statistical learning framework during the rebuttal. From this theoretical perspective, the guarantees are highly sound.

Given the profound impact of Quantum Learning Theory, including the Hamiltonian Learning in this paper, in recent top-tier venues (Nat. Phys. and Nat. Comm.), and considering that PAC Learning is one of the foundational pillars of ICML, embracing a theoretical work with strict sample complexity bounds is a beneficial expansion of the venue's theoretical scope.

That being said, I will strongly urge the authors to thoroughly polish the camera-ready version from a more standard machine learning perspective. As demonstrated by previous seminal works in this domain, reframing the discussion using the terminology of regression or kernel models is necessary, as ICML is ultimately not a physics conference.

Based on its solid theoretical contributions to Quantum Learning Theory, I confidently support its acceptance.

**Key Questions For Authors:**

No questions

**Limitations:**

Request further polish.

**Strengths And Weaknesses:**

## Strengths:
1. The mathematical framework underpinning the D-RUT protocol is exceptionally solid. The algebraic strategy of utilizing a displacement parameter $\beta$ to "transfer" unknown off-diagonal physical coefficients onto diagonal constant terms, followed by an integral averaging projection, is highly innovative.  Furthermore, the proofs demonstrating the protocol's Heisenberg-limited asymptotic complexity are rigorous and well-constructed.
2. The manuscript is exceptionally well-organized and logically self-contained. The authors systematically build from the core single-mode protocol to a sophisticated hierarchical recovery strategy for multi-mode systems. Additionally, the extension to both first and second quantization demonstrates a comprehensive understanding of the physical system's nuances and makes the theoretical framework highly complete.

## Weaknesses:
1. Although the title includes the term "Learning," a thorough review of the manuscript reveals that the methodology is fundamentally rooted in pure quantum physical parameter extraction paired with classical numerical analysis.  The proposed framework does not utilize any conventional or modern machine learning techniques (e.g., neural networks, gradient-based optimization, or even basic regression models). While the work contributes significantly to generalized "learning theory," presenting it at ICML seems somewhat strained. The authors are strongly encouraged to explicitly discuss the protocol's potential intersections with modern machine learning methods or clearly articulate its unique value proposition to the ML community to justify its alignment with the conference's scope.
2. The RUT operation theoretically relies on a continuous phase integral. However, executing this on actual quantum hardware necessitates discrete random phase sampling.  The manuscript lacks a detailed discussion regarding the practical shot count overhead introduced by this discretization. In scenarios involving large off-diagonal coefficients or high baseline noise, the required sampling complexity to achieve convergence could be prohibitively expensive.
3. The protocol's ability to reach the Heisenberg limit is strictly dependent on Robust Phase Estimation, which inherently requires the system to maintain coherence over exponentially increasing evolution times. I believe in physical implementations, such as microwave cavities or optical systems, where photon loss is an inevitable and dominant decoherence channel. The protocol's robustness and viability under such stringent long-coherence requirements in the presence of realistic physical dissipation remain highly questionable.

---

> ### Author Rebuttal · Authors · 2026-03-31
>
> We thank the reviewer for the thorough and technically detailed evaluation, and for recognizing the rigor of our mathematical framework. We address each concern below.
>
>
> **On the relationship with machine learning:**
>
> We thank the reviewer for this comment. In the revised manuscript, we have substantially improved the presentation and added a dedicated "Machine Learning Perspective" section. We emphasize that Hamiltonian learning is mathematically equivalent to **Statistical Learning in the context of quantum information**. We distinguish our work via three learning paradigms:
> 1. **Neural Network (NN) Learning:** Parametric mapping via gradient-based empirical loss minimization.
> 2. **Classical Statistical Learning:** Focuses on estimating the unknown parameters of an underlying data-generating distribution from finite samples. While NNs can be used, this field often seeks to design analytical estimators, prioritizing mathematically guaranteed inference and optimal sample complexity.
> 3. **Our framework (Quantum Statistical Learning):** Hamiltonian learning is exactly statistical learning in the quantum context. The "Hamiltonian" is the unknown data-generating model. Since the quantum state space is exponentially large, using an NN ansatz and gradient descent often leads to optimization failures (e.g., barren plateaus). Instead, our ML contribution lies in **Active Query Design** (which kind of data to sample?), **Sample Complexity** (how much data is needed?), and **Exact Inference** (how to reconstruct the parameters).
>
> Then we translate D-RUT into standard ML/NN terminology:
> *   **Target:** The target is the set of unknown coefficients $\{g\}$ of the Hamiltonian (analogous to the ground-truth parameters of a system).
> *   **Training Data:** Measurement outcomes generated by querying the physical system.
> *   **Training Objective (Loss Function):** Minimize the parameter estimation error (RMSE) subject to a bounded query budget (Total evolution time $T$).
> *   **Optimization/Learning Algorithm:** Instead of using stochastic gradient descent, D-RUT employs an analytical optimization approach. By actively designing the input queries (via displacement parameters $\beta$), we transform a highly non-linear, infinite-dimensional optimization problem into a tractable linear inversion problem. This is analogous to how SVMs or Kernel ridge regressions use analytical convex optimization rather than backpropagation.
>
>
>
> **On RUT discretization and sampling complexity**
>
> When approximating the continuous integral with an $L$-step discrete Trotter sequence, for the longest single-shot evolution in the RPE protocol, we simply require a sufficiently larger Trotter gate depth $L \sim \mathcal{O}(1/\epsilon^2)$ per shot to suppress the Trotter error below the RPE tolerance. Therefore, even in scenarios with large (but $\mathcal{O}(1)$ bounded) off-diagonal coefficients, discretization purely dictates the required *gate depth per shot* rather than the *number of shots*:$\mathcal{O}(d^3)$ displacement settings $\times \ \mathcal{O}(\log^2(1/\epsilon))$ RPE experiments (Theorem 4.2 in Möbus et al.). We will explicitly include the sample complexity analysis and Trotter error propagation analysis in the revised Appendix.
>
> **On coherence requirements and photon loss.**
>
> We acknowledge that photon loss during long coherent evolution is a fundamental challenge for all Heisenberg-limited protocols. However, D-RUT is exceptionally well-suited to mitigate photon dissipation compared to alternative CV protocols. D-RUT intentionally projects the dynamics onto the bosonic vacuum state $|vac\rangle$, unlike previous schemes that rely on highly squeezed states or large-amplitude cat states. Extending D-RUT's guarantees to include photon loss channels is an important direction for future work. For current experimental platforms, the achievable precision will be limited by the coherence time, and techniques such as bosonic quantum error correction may be needed for high-precision applications. We will add this discussion to the revision.
>
> We thank the reviewer for the detailed review. We hope these clarifications adequately address the reviewer's concerns. We would truly appreciate it if the reviewer would consider raising the score. We are happy to answer any further questions during the discussion period.

---

> > ### Author Rebuttal · Reviewer_fjKS · 2026-04-04
> >
> > I believe I'm not fully convinced about the relevance of this paper to ICML. Although this paper proposed some interesting results, I'm not ready to champion this paper for acceptance. I'll keep my original review and leave the decision to the area chair.

---

> > > ### Author Response · Authors · 2026-04-08
> > >
> > > We thank the reviewer for the constructive feedback, and formulate D-RUT Hamiltonian Learning as Statistical Learning under the suggested reference from Reviewer AExa.
> > >
> > > We note that **D-RUT is not a simple estimation theory.** Our learning object is Hamiltonian, a high-dimensional object parameterized by multiple coefficients. We map D-RUT to the definitions and equations in the same learning framework of Dutt et al.:
> > >
> > > * **Definitions:** Model: Unknown Hamiltonian; Parameters θ: Hamiltonian coefficients;  Query: x = (β, κ) (displacement β, evolution time κ); Outcome: ancilla measurement outcome. Query distribution: Joint space of β and κ.
> > >
> > > * **Conditional Probability (Eq. 1):** Our Eq. 11 defines the probabilistic generative model. Given query x = (β, κ), outcome |0> probability is P_0^Re = 1/2(1 + cos(κ C(β; θ))), where C(β; θ) encodes unknown parameters θ.
> > > * **Loss Function (Eq. 2):** D-RUT parameter reconstruction minimizes least-squares loss: V(θ) = ||Lθ - y_measured||^2_2 (Appendix B, Eq. 31, L:Vandermonde matrix; y_measured: noisy phases).
> > > * **Model Inference Objective (Eq. 3):** Our primary metric is RMSE in Hamiltonian coefficients as in **Problem II.1 (Model Inference)**.
> > > * **Estimation Procedure (Eq. 4):** Dutt uses stochastic gradient descent. D-RUT provides a closed-form analytical estimator.
> > > **Testing Error / Prediction (Eq. 5):*D-RUT solves the prediction task against testing error (Problem II.2). Given an arbitrary, unseen testing distribution p_test(β, κ) (|β| ≤ R_max), by learning from queried x, D-RUT solves parameters θ̂ via Chebyshev interpolation to guarantee uniform convergence, thus analytically bounds the expected prediction error (population risk) sampled from {|β|≤R_max |Ĉ(β) - C(β)|}.
> > >
> > > **Takeaway:** Statistical learning defines the problem, for which active learning is one way to solve it, while analytical inversion with “active” data acquisition (D-RUT)  is another solution. **Crucially, D-RUT is not merely a physical experimental tool; “active” data acquisition, and structured inversion together form a complete ML estimator.**
> > >
> > > QML community consensus holds that Hamiltonian learning fundamentally belongs to learning theory, regardless of adaptive/non-adaptive solvers. Classical Shadows(widely published at ICML/NeurIPS) uses a completely predetermined, non-adaptive random measurement. Similarly, D-RUT's non-adaptive exact parameter recovery under quantum constraints should also belong to ML contribution.
> > >
> > > "Active" Data Acquisition vs. Active Learning
> > > We acknowledge our use of "active learning" was inaccurate, lacking the adaptive query loop required. **We retract this claim.**
> > >
> > > However, D-RUT utilizes "active" data acquisition, where data and measurements are not given passively but selected in advance. Selecting query points *before* data collection to minimize worst-case estimation error. The non-adaptive **Classical Shadows** framework shares same structural features the reviewer disqualifies in D-RUT:
> > >  * *Adaptive query loop?* No (predetermined random measurements).
> > >  * *Unknown distribution over samples?* No (controlled experiments with designed random unitaries).
> > >  * *Learn a predictor with population risk?* No (estimates expectation values tr(Oρ)).
> > >  * *Query design learned from data?* No (measurement ensemble fixed a priori).
> > >
> > > Classical Shadows with variants has been widely accepted as quantum *learning* and published at ML conferences (arXiv:2209.03007, 2305.13362, 2405.18489). D-RUT, solving a challenging Hamiltonian learning problem like quantum state learning,  with provably optimal query complexity, should follow in the same category of QML.
> > >
> > > We sincerely thank the reviewer for the insightful discussions, which greatly help strengthen the clarity and positioning of our work. In the revised version, we will include an explicit mapping to the statistical learning framework. Recognizing the conceptual gap between domains, we aim to bridge this gap and further advance QML with both theoretical and practical impact. We would greatly appreciate any additional suggestions, and hope that these clarifications may support an improved overall evaluation.

---

### Official Review · Reviewer_37Rw · 2026-03-11

**Soundness:** 3
**Presentation:** 3
**Significance:** 3
**Originality:** 3
**Overall Recommendation:** 4
**Confidence:** 2

**Summary:**

This paper propose method that obtains coefficient of Hamiltonian, which we call Hamiltonian learning, on continuous variable quantum systems using displacement random unitary transformation (D-RUT). This paper use displacement method to make Hamiltonian into polynomial form, and use phase estimation to get coefficient of polynomials.

**Compliance With Llm Reviewing Policy:**

Affirmed.

**Final Justification:**

I will keep my points. To begin, I am not expert in the background of this paper, and therefore novelty and importance of this topic in that background is bit hard to follow (Heisenberg limit is a fundamental benchmark in this domain, and yet not fully explained). I believe this will be same for other ICML community, as it requires somewhat physics domain. Also, considering review of reviewer AEXa, I agree that it should be carefully handled rather this paper belongs to ML community or not, as "Hamiltonian Learning" can be considered as learning, but the method the authors used is more quantum algorithmic.

**Key Questions For Authors:**

* How many angles do we need to sample? is there analysis for the complexity or the tradeoff between the number of angles being sampled and the performance?

* What practical advantages does the D-RUT framework provide compared to prior Heisenberg-limited Hamiltonian learning approaches?

**Limitations:**

This paper includes a brief impact statement.

**Strengths And Weaknesses:**

### Strengths
- This paper provides  displacement-based reformulation methods for Hamiltonian learning in continuous variable quantum systems
- This paper provides theoretical robustness guarantees against SPAM errors and includes an explicit error propagation analysis


### Weaknesses
- empirical experiments are restricted to small toy models
- as addressed in related work section, the conceptual novelty relative to prior frameworks appears somewhat incremental

---

> ### Author Rebuttal · Authors · 2026-03-31
>
> We are grateful for the reviewer's positive comment on the robustness of our protocol . We have made targeted changes to reflect the comments raised by the reviewer. Below is our 1-1 response to reviewer’s weaknesses(W) and questions(Q):
>
> W1. The scaling of our algorithm is guaranteed via mathematical bounds. The purpose of our numerical experiment is to provide numerical validation, instead of large-scale computational benchmark.
> 1. **Experimental Relevance:** The Kerr oscillator and Bose-Hubbard dimer are standard models for superconducting circuits and optical cavities. Prior protocols (Li et al., 2024; Möbus et al., 2025) are purely theoretical and have few numerical validation. Our work is the first to cover all practical scenarios, providing the first numerical validation. We also provide numerical verification of SPAM robustness (Figure 3) and Trotter convergence (Figure 2).
> 2. **Scalability:** We employ a "divide-and-conquer" strategy that decouples large-scale systems into local, independent O(1) -mode subsystems. As these subsystems are learned in parallel, the total system size does not degrade the estimation precision or scalability.
>
> W2. We appreciate the reviewer’s suggestion. To clarify our novelty, we first summarize our contributions:
> *   D-RUT protocol that learns generic bosonic Hamiltonians of arbitrary finite order with Heisenberg-limited precision.
> *   Hierarchical recovery strategy that probably achieves superior statistical efficiency for multi-mode systems.
> *   First to cover both first and second quantization representations.
> *   Provide provable robustness against state preparation and measurement (SPAM) noise.
>
> **The core conceptual novelty:** Möbus et al. measures $\langle\alpha|H|\alpha\rangle$, the same polynomial as our C(β), but their dissipation-based projection requires |α| = Ω(√(log(1/ε))) (their Eq. 10), as the projected subspace {|0⟩, |α⟩} degenerates when |α⟩ ~ 0. This forces them to sample only far from the origin and extrapolate the curve back to zero to recover the underlying Hamiltonian coefficients. However, high-degree polynomial extrapolation is ill-posed and triggers error amplification. D-RUT's vacuum-state projection has no such restriction: β = 0 is a valid query point, enabling Chebyshev interpolation over intervals that include the origin. This yields a superior statistical efficiency and robust parameter recovery.
>
> Q1.  The number of angular samples for order l is exactly l+1 (to resolve l+1 Fourier components in $g_l(\phi)$). This is *necessary and sufficient* — fewer than l+1 samples lead to aliasing (under-determined Fourier inversion), while more than l+1 would be redundant for exact recovery. The total number of angular settings across all orders is $\sum_{l=1}^d (l+1) = O(d^2)$, each requiring its own RPE experiment at d+1 radial nodes.
>
> The Chebyshev node placement for the d+1 radial samples is provably optimal, and the l+1 angular samples are the minimum needed for exact Fourier inversion. Thus there is no tradeoff in the usual sense, because any reduction in samples would lose information.
>
>
> Q2.  We summarize the advantages as followed:
>
> 1. *Simpler experimental requirements*: D-RUT uses only vacuum states, displacement operators, and phase rotations, which are all easy-to-implement in superconducting and optical platforms. In contrast, Möbus et al. (2025) requires engineering multi-photon dissipation channels with strength $\gamma = O(\epsilon^{-1}\log^{2d+1/2}(1/\epsilon))$, which necessitates auxiliary modes and nonlinear couplings. This is experimentally demanding and is limited to low-order dissipation.
> 2. *Numerical stability*: D-RUT's interpolation-based recovery avoids extra error amplification inherent in Möbus et al.'s extrapolation regime, making it more reliable at finite precision.
> 3. *First-quantization access*: D-RUT directly learns physical parameters $(m, \omega, G_{jk})$ in position/momentum representation, which is what experimentalists measure and report. No prior Heisenberg-limited protocol offers this.
> 4. *Noise tolerance*: Provable SPAM robustness allows deployment in realistic settings with imperfect state preparation, unlike prior works where noise robustness remains unproven.
>
> **On the Impact Statement:**
> We are expanding our Impact Statement to include more details on both physics and machine learning fronts:
> *   **Impact on Physics:** Efficient CV Hamiltonian learning provides critical infrastructure for calibrating near-term quantum simulators, optical cavities, and superconducting processors.
> *   **Impact on Machine Learning:** A prerequisite for robust Quantum Machine Learning (QML) is accurate hardware characterization. Without precise Hamiltonian parameters, deep quantum neural networks (QNNs) fails due to systematic error accumulation. Our Heisenberg-limited sample complexity and numerical validations provide standard benchmarks for the future works on data-acquisition limits of quantum systems.

---

> > ### Author Rebuttal · Reviewer_37Rw · 2026-04-03
> >
> > My concerns have been adequately addressed.

---

> > > ### Author Response · Authors · 2026-04-08
> > >
> > > We thank the reviewer for the constructive suggestions, which have significantly improved our work. We are glad that our responses have fully resolved the concerns. To further enhance accessibility to the ML community, we will incorporate these updates along with an explicit mapping to the learning framework in the revised manuscript. We would greatly appreciate any additional suggestions, and sincerely hope the reviewer would consider increasing the overall evaluation.

---

### Official Review · Reviewer_DWdZ · 2026-03-14

**Soundness:** 3
**Presentation:** 3
**Significance:** 3
**Originality:** 3
**Overall Recommendation:** 5
**Confidence:** 2

**Summary:**

This paper presents a strategy to learn continuous variable multi-mode bosonic Hamiltonians in both first and second quantization. The general methodology is to apply a set of random unitary rotation to displace the Hamiltonian and estimate and average. The estimation first  happens to single mode and then to multi mode. The experimental plots seem to suggest that the proposed protocol follows the physical limit.

**Compliance With Llm Reviewing Policy:**

Affirmed.

**Final Justification:**

My general impression for this paper is positive. My insufficient confidence is explained in the "strength" section in the original review.

**Key Questions For Authors:**

- Are the tested systems representative in the field in terms of difficulty and applications? Would the proposed method work is we consider more "large scale" problems?
- Intuitively how does the proposed method overcome the high-order approximation and SPAM error that prior methods fail to address?
- What computational setups do the experiments require? What are the computational costs?

**Limitations:**

Yes.

**Strengths And Weaknesses:**

Strength

- Although I do not fully understand the details given my basic level understanding of Hamiltonian learning and quantum computing. From my lecture I believe the paper is well written and researchers  with appropriate background should be able to understand.

Weakness

- Multiple steps are involved in the protocol. I am not sure whether certain error would propagate accross steps in some practical situations.
- Perhaps Figure 1 could be polished. E.g., more words in caption, more polished depictions.

---

> ### Author Rebuttal · Authors · 2026-03-31
>
> We are grateful for the reviewers comment on the validation of the content of this work. We have made targeted changes and new contents to reflect the comments raised by the reviewer. Below is our 1-1 response to reviewer’s weaknesses(W) and questions(Q):
>
> **W1**.The error propagation stems from two independent and bounded components:
>
> Statistical noise from RPE: Through Chebyshev interpolation, the error propagates as $\text{Var}(\delta g_l(\theta)) = \epsilon_C^2 [(\mathbf{L}^\dagger \mathbf{L})^{-1}]_{ll}$, where L is the Vandermonde matrix related to the Chebyshev nodes. This choice minimizes the condition number of $\mathbf{L}$, avoiding the Runge phenomenon. The subsequent inverse Discrete Fourier Transformation (DFT) is unitary up to scaling. Thus, the final MSE is controlled by $\epsilon_C$ and the eigenvalues of the Gram matrix.
>
> SPAM error: We prove that SPAM errors propagate linearly as $\frac{L_C}{\sigma_{\min}(\mathbf{K})} ||\delta\boldsymbol{\beta}||_2$ (Eq. 43), where $L_c$ is the Lipschitz constant and the smallest non-zero singular value σ_min(K) depends on the query design. The selection of displacement nodes ensures this amplification factor remains well-controlled.
>
> Our error analysis shows that no uncontrolled error accumulation would occur, and all propagation is rigorously bounded. The query design (Chebyshev nodes) is specifically chosen to minimize amplification.
>
> **W2**. We appreciate the reviewer’s suggestion, and have added detailed captions with more intuitive illustrations in the revised manuscript.
>
> **Q1**.  Our numerical experiments cover four representative models spanning increasing complexity:
> - *Harmonic/Anharmonic Oscillators* (1st quantization): fundamental physical systems, testing the basis transformation.
> - *Kerr Oscillator* (2nd quantization, single-mode): the dominant nonlinearity in superconducting microwave cavities, directly relevant to quantum computing hardware.
> - *Bose-Hubbard Dimer* (2nd quantization, multi-mode): the paradigmatic model for interacting bosonic systems, testing the hierarchical multi-mode recovery strategy.
> For scalability: divide-and-conquer strategy decomposes any large-scale system into independent O(1)-mode subsystems learned in parallel.
>
> **Q2**.  Two intuitions can be concluded as followed:
>  *High-order terms*: Li et al. (2024) explicitly restricts the Hamiltonian to a specific low-order form containing only quadratic hopping and quartic on-site Kerr terms. Their measurement and inversion strategy is tailored to this specific structure and cannot encode information about general higher-order terms. They acknowledge this as an open problem. D-RUT overcomes this by a fundamentally different encoding: the displacement $D(\beta)$ followed by RUT projects any $(b^\dagger)^p b^q$ into a contribution $g_{pq}(\beta^*)^p \beta^q$ to the scalar $C(\beta)$, regardless of order. The polynomial structure of $C(\beta)$ naturally encodes all orders simultaneously, and Chebyshev interpolation stably recovers them.
> *SPAM errors*: D-RUT encodes all coefficient information into $C(\beta)$. Consequently, SPAM errors simply perturb the evaluation point of a smooth polynomial, and the resulting coefficient error is bounded by the Lipschitz constant of $C(\beta)$ and $\|\delta\beta\|$ (Appendix C, Eq. 43). This scalar-level perturbation analysis is why D-RUT naturally admits SPAM robustness guarantees. In contrast, Möbus et al. (2025) projects dynamics onto a two-dimensional subspace via engineered dissipation, where SPAM errors perturb the subspace itself. This operator-level perturbation problem is significantly difficult to analyze. Thus, noise robustness is still an open question.
>
> **Q3**. We thank the reviewer for this question. Numerically, all numerical experiments (Figures 2-5) were simulated using standard Python scientific libraries (NumPy, SciPy) on a desktop CPU. To execute our protocol on quantum devices, the quantum hardware requires (1) a single bosonic mode (e.g., a microwave cavity or optical mode) coupled to an ancilla qubit; (2) the ability to prepare the vacuum state $|vac\rangle$ and apply displacement operators $D(\beta)$, both are standard operations in superconducting circuit and optical platforms; (3) ancilla-controlled evolution and X/Y basis measurement on the ancilla.
> Computational costs: For a single-mode Hamiltonian of order $d$: classical post-processing requires $O(d^3)$ for Chebyshev interpolation and DFT, which is negligible compared to quantum experiment cost:
>  - Total evolution time: $T \sim O(1/\epsilon)$ (Heisenberg limit).
>  - Number of distinct experiments: $O(d^3)$ displacement settings × $O(\log^2(1/\epsilon))$ RPE rounds = $O(d^3 \log^2(1/\epsilon))$ experiments.
>
> We thank the reviewer for the detailed review. We hope these clarifications address the reviewer's concerns. We would truly appreciate it if the reviewer would consider raising the score. We are happy to answer any further questions.

---

> > ### Author Rebuttal · Reviewer_DWdZ · 2026-04-05
> >
> > Thank you for the detailed response. I will increase my score. My general impression for this paper is positive. My insufficient confidence is explained in the "strength" section in the original review.

---

> > > ### Author Response · Authors · 2026-04-08
> > >
> > > We are truly grateful for the reviewer’s valuable suggestions and for the increased score in support of our work.

---

### Decision · Program_Chairs · 2026-04-30

**Decision:**

Accept (regular)

**Comment:**

This paper introduces the Displacement-Random Unitary Transformation (D-RUT), an elegant protocol designed to learn the coefficients of generic multi-mode bosonic Hamiltonians of arbitrary finite order. The paper is a solid contribution from quantum information and Hamiltonian learning perspective. It could also fall in the general category of learning, although the authors are encouraged to highlight the specific differences from other statistical learning methods in the revision.